# Learning Thresholds with Latent Values and Censored Feedback

**Jiahao Zhang**[1,2], **Tao Lin**[3], **Weiqiang Zheng**[4], **Zhe Feng**[5], **Yifeng Teng**[5], **Xiaotie Deng**[1,2]

[1]School of Electronics Engineering and Computer Science, Peking University

[2]CFCS, Peking University  [3]Harvard University  [4]Yale University  [5]Google Research

`{jiahao.zhang, xiaotie}@pku.edu.cn, tlin@g.harvard.edu,`
`weiqiang.zheng@yale.edu, {zhef, yifengt}@google.com`

## Abstract

In this paper, we investigate a problem of *actively* learning threshold in latent space, where the *unknown* reward $g(\gamma, v)$ depends on the proposed threshold $\gamma$ and latent value $v$ and it can be achieved *only* if the threshold is lower than or equal to the *unknown* latent value. This problem has broad applications in practical scenarios, e.g., reserve price optimization in online auctions, online task assignments in crowdsourcing, setting recruiting bars in hiring, etc. We first characterize the query complexity of learning a threshold with the expected reward at most $\varepsilon$ smaller than the optimum and prove that the number of queries needed can be infinitely large even when $g(\gamma, v)$ is monotone with respect to both $\gamma$ and $v$. On the positive side, we provide a tight query complexity $\tilde{\Theta}(1/\varepsilon^3)$ when $g$ is monotone and the CDF of value distribution is Lipschitz. Moreover, we show that a tight $\tilde{\Theta}(1/\varepsilon^3)$ query complexity can be achieved as long as $g$ satisfies right Lipschitzness, which provides a complete characterization for this problem. Finally, we extend this model to an online learning setting and demonstrate a tight $\Theta(T^{2/3})$ regret bound using the aforementioned query complexity results.

## 1 Introduction

The thresholding strategy is widely used in practice, e.g., setting bars in the hiring process, setting reserve prices in online auctions, and setting requirements for online tasks in crowdsourcing, due to its intrinsic simplicity and transparency. In addition, in the practical scenarios mentioned above, the threshold can be only set in a *latent* space, which makes the problem more challenging:

- In hiring, the recruiter wants to set a recruiting bar that maximizes the quality of the hires without knowing the true qualifications of each candidate.

- In online auctions, the seller wants to set a reserve price that maximizes their revenue. However, the seller does not know the true value of the item being auctioned.

- In crowdsourcing, the taskmaster wants to set a requirement (e.g., the number of questions that need to be answered) for each task so that the tasks can be assigned to qualified workers who have a strong willingness to complete the task, but has no access to the willingness of each worker. The target of the taskmaster is to collect high-quality data as much as possible.

Although the latent value cannot be observed, it will affect the reward jointly with the threshold. In the examples above, the reward is the quality of the hire, revenue from the auction, or the quality of the collected data. In some cases, the reward only depends on the latent value as long as the value is larger than the threshold, e.g., the quality of the hire may only depend on the intrinsic qualification and skill level of the candidate as long as she passes the recruiting bar, the revenue in second price auction only depends on the value since the bidders will report truthfully regardless of the reserve price. However, the threshold does play an important role in other settings, e.g., the winning bid in

the first price auction relies on and increases with the reserve price in general (Krishna, 2010) thus the revenue highly depends on both reserve price and latent value. Similarly, in crowdsourcing, the requirements of the tasks directly affect the quality of data collected by the taskmaster. In this work, we consider a general framework where the reward can depend on both threshold and latent value.

Another difficulty in setting a proper threshold is to balance the per-entry reward and capability. For example, in hiring, setting a higher recruiting bar can guarantee the qualification of each candidate and thus the return to the company. However, a too-high bar may reject all candidates and cannot fit the position needs. To balance this tradeoff, the threshold needs to be set appropriately.

## 1.1 OUR MODEL AND CONTRIBUTION

**An informal version of our model** In this work, we propose a general active learning abstraction for the aforementioned thresholding problem. We first provide some notations of the model considered in this paper to facilitate the presentation. Let $g(\gamma, v)$ be the reward function that maps the threshold $\gamma \in [0, 1]$ and latent value $v \in [0, 1]$ to the reward, where $g(\gamma, v)$ can be observed if and only if $\gamma \leq v$. We assume that the latent value follows an *unknown* distribution and we can only observe the reward (as long as $\gamma \leq v$) but not the value itself. We investigate the query complexity of learning the optimal threshold in the latent space: the number of queries that are needed to learn a threshold with an expected reward at most $\varepsilon$ smaller than the optimum.

**Our contributions** The first contribution in this work is an impossibility result for this active learning problem. We prove that the query complexity can be infinitely large *even* when the reward function is monotone with respect to the threshold and the value. Our technique is built upon the idea of "needle in the haystack". Intuitively, a higher threshold $\gamma$ gives a higher reward $g(\gamma, v)$ because of the monotonicity but decreases the probability of getting a reward which can only be achieved if $\gamma \leq v$. This tradeoff allows us to construct an interval that has an equal expected utility. We find that if the reward function has a discontinuous bump at some point in the interval and the value distribution has a mass exactly at the same point, then the expected utility at this point will be constantly higher than the equal expected utility. Therefore we can hide the unique maximum ("needle") in an equal utility interval ("haystack"), which makes the learner need infinite queries to learn the optimal threshold.

Our second contribution is a series of positive results with *tight* query complexity up to logarithmic factors. We consider two special cases with common and minor assumptions: (1) the reward function is monotone and the CDF of value distribution is Lipschitz and (2) the reward function is right-Lipschitz. With each of the two assumptions, we can apply a discretization technique to prove an $\tilde{O}(\frac{1}{\varepsilon^3})$ upper bound on the query complexity of the threshold learning problem. We also give a matching lower bound, which is technically more challenging: at least $\Omega(\frac{1}{\varepsilon^3})$ queries are needed to find an $\varepsilon$-optimal threshold, even if the reward function is both monotone and Lipschitz and the value distribution CDF is Lipschitz. Previous papers like Kleinberg & Leighton (2003); Leme et al. (2023b) do not require the value distribution to be Lipschitz, which makes it much easier to construct a value distribution with point mass to prove lower bounds. To prove our lower bound with strong constraints on the reward function and distribution, we provide a novel construction of value distribution by careful perturbation of a smooth distribution. We summarize our main results in Table 1.

Finally, we extend the threshold learning problem to an online learning setting. Relating this problem to continuous-armed Lipschitz bandit, and using the aforementioned query complexity lower bound, we show a tight $\Theta(T^{2/3})$ regret bound for the online learning problem.

## 1.2 RELATED WORKS

The most related work is the pricing query complexity of revenue maximization by Leme et al. (2023b;a). They consider the problem of how many queries are required to find an approximately optimal reserve price in the posted pricing mechanism, which is a strictly special case of our model by assuming $g(\gamma, v) = \gamma$. Our work also relates to previous work about reserve price optimization in online learning settings, e.g., Cesa-Bianchi et al. (2014); Feng et al. (2021), who consider revenue maximization in the online learning setting, where the learner can control the reserve price at each

Table 1: Our results on the query complexity of learning optimal threshold. Rows correspond to reward functions and columns to value distribution. $\tilde{O}(\cdot)$ ignores poly-logarithmic factors.

| Reward / Value | Lipschitz | | General | |
|---|---|---|---|---|
| | lower bound | upper bound | lower bound | upper bound |
| Monotone | $\Omega(\frac{1}{\varepsilon^3})$ Theorem 4.3 | $\tilde{O}(\frac{1}{\varepsilon^3})$ Theorem 4.1, 4.2 | Infinite Theorem 3.1 | |
| Right-Lipschitz | | | $\Omega(\frac{1}{\varepsilon^3})$ Theorem 4.3 | $\tilde{O}(\frac{1}{\varepsilon^3})$ Theorem 4.2 |

round. Our work is loosely related to the sample complexity of revenue maximization (e.g., (Cole & Roughgarden, 2014; Morgenstern & Roughgarden, 2015; Gonczarowski & Nisan, 2017; Guo et al., 2019; Brustle et al., 2020)). They focus on learning near-optimal mechanisms, which lies in the PAC learning style framework. Whereas, our work characterizes the query complexity in the active learning scenario.

Technically, our work is inspired by a high-level idea called "the needle in the haystack", which was first proposed by Auer et al. (1995) and also occurred in recent works such as online learning about bilateral trade (Cesa-Bianchi et al., 2021; 2023a), first-price auction (Cesa-Bianchi et al., 2023b), and graph feedback (Eldowa et al., 2023). Nevertheless, this idea is only high-level. As we will show in the proofs, adopting this idea to prove our impossibility results and lower bounds is not straightforward and requires careful constructions.

Additionally, our work is related to works on censored/truncated data and the Lipschitz bandit problem. We leave them to the Appendix A.

## 2 MODEL

We first define some notations. For an integer $n$, $[n]$ denotes the set $\{1, 2, ..., n\}$. $\mathbf{1}_{(\cdot)}$ is the indicator function. We slightly abuse notation to use $F$ to denote both a distribution and its cumulative distribution function (CDF).

We study the query complexity of learning thresholds with latent values between a *learner* and an *agent*. The *latent value* $v$ represents the agent's private value and is drawn from an *unknown* distribution $F$ supported on $[0, 1]$. In each query, the learner is allowed to choose a *threshold* $\gamma \in [0, 1]$; then with a fresh sample $v \sim F$, the learner gets reward feedback $b$ determined by the threshold $\gamma$ and the value $v$:

$$b(\gamma, v) = \begin{cases} g(\gamma, v) & \text{if } v \geq \gamma \\ 0 & \text{if } v < \gamma \end{cases} \tag{1}$$

where $g : [0, 1]^2 \to [0, 1]$ is an *unknown reward function*. The notation $b(\gamma) = b(\gamma, v)$ denotes a random reward with randomness $v \sim F$. The learner's goal is to learn an optimal threshold $\gamma^* \in [0, 1]$ that maximizes its expected reward/utility $U(\gamma)$ defined as

$$U(\gamma) \triangleq \mathbb{E}_{v \sim F}\big[b(\gamma, v)\big] = \mathbb{E}_{v \sim F}\big[g(\gamma, v) \cdot \mathbf{1}_{v \geq \gamma}\big]. \tag{2}$$

Typically, a higher threshold decreases the probability of getting a reward but gives a higher reward if the value exceeds the threshold. Our model of learning optimal thresholds with latent value captures many interesting questions, as illustrated by the following examples.

**Example 2.1** (reserve price optimization). *A seller (learner) repeatedly sells a single item to a set of $n$ bidders. The seller first sets a reserve price (threshold) $\gamma$. Each bidder $i$ then submits a bid $b_i$. The bidder with the highest bid larger than $\gamma$ wins the item and pays their bid; if no bidder bids above $\gamma$, the item goes unallocated. Each bidder $i$ has a private valuation $v_i \in [0, 1]$ for the item, where each value $v_i$ is drawn independently (but not necessarily identically) from some unknown distribution. If the seller adopts the first-price auction, only the highest bid matters for both allocation and payment. We denote the maximum value (latent value) $v^{(1)} \triangleq \max v_i$ and $v^{(1)}*

*is drawn from a unknown distribution. Then we only consider this representative highest bidder (agent).[1] The unknown bidding function (reward function) $g$ and $g(\gamma, v^{(1)})$ is the maximum bid when the reserve price is $\gamma$ and the maximum value is $v^{(1)}$. If the seller adopts the second-price auction, only the second-highest value matters for both allocation and payment. We denote the second highest value (latent value) $v^{(2)}$ and $v^{(2)}$ is drawn from an unknown distribution. Similarly, we only need to consider the second-highest bidder (agent). Because bidders in the second price auction bid truthfully, it has a bidding function (reward function) $g$ with $g(\gamma, v^{(2)}) = v^{(2)}$ for all $\gamma, v^{(2)} \in [0, 1]$ and $v^{(2)} \geq \gamma$. If the seller faces a single bidder (agent) and adopts a posted price auction, we have $g(\gamma, v) = \gamma$ as the bid when the reserve price is $\gamma$ and the bidder's value is $v$.*

**Example 2.2** (crowdsourced data collection). *Data crowdsourcing platforms typically allow users (agent) to sign up and complete tasks in exchange for compensation. These tasks might involve answering questions, providing feedback, or rating products. The taskmasters (learner) need to decide how many questions should be included in each task or how detailed the feedback should be. We denote such difficulties (threshold) of the task $\gamma$. The users have individual willingness (latent value) $v$ to complete the task. The willingness follows a specific probability distribution, which is unknown to the taskmasters or platform. Typically, a more difficult task decreases the probability of getting feedback from the users but gives a higher reward if the users are willing to complete the tasks because more detailed information can be included when using a more difficult task. We use the notation $g(\gamma, v)$ to represent the reward when the difficulty of the task is $\gamma$ and the user's willingness is $v$.*

**Example 2.3** (hiring bar). *A company (learner) sets a predefined bar (threshold) $\gamma$ for candidate admission. These candidates (agents) have individual measurements (latent value) $v$, which reflects their inherent ability. They will apply to the company if and only if they think of themselves as qualified, namely, $v \geq \gamma$. The measurements follow a specific probability distribution, which is unknown to the company. A candidate with a measurement $v$ admitted with a hiring bar $\gamma$ produces an output (reward) $g(\gamma, v)$ to the company.*

We assume that the value distribution $F$ belongs to some class $\mathcal{C}$, and the reward function $g$ belongs to some class $\mathcal{G}$. The classes $\mathcal{C}$ and $\mathcal{G}$ are known to the learner. The learner makes $m$ queries adaptively and then outputs a threshold $\hat{\gamma} \in [0, 1]$ according to some algorithm $\mathcal{A}$, namely, $\gamma_t = \mathcal{A}(\gamma_1, b_1, \ldots, \gamma_{t-1}, b_{t-1})$, $b_t = b(\gamma_t, v_t), \forall t \in [m]$, and $\hat{\gamma} = \mathcal{A}(\gamma_1, b_1, \ldots, \gamma_m, b_m)$.

**Definition 2.1** (($\varepsilon, \delta$)-estimator). *An ($\varepsilon, \delta$)-estimator (for $\mathcal{C}$ and $\mathcal{G}$) is an algorithm $\mathcal{A}$ that, for any $F \in \mathcal{C}, g \in \mathcal{G}$, can output a $\hat{\gamma}$ satisfying $U(\hat{\gamma}) \geq U(\gamma^*) - \varepsilon$ with probability at least $1 - \delta$ using $m$ queries (where the randomness is from $v_1, \ldots, v_m \sim F$ and the internal randomness of $\mathcal{A}$).*

**Definition 2.2** (query complexity). *Given $\mathcal{C}, \mathcal{G}$, for any $\varepsilon > 0$ and $\delta \in (0, 1)$, the query complexity $\mathrm{QC}_{\mathcal{C},\mathcal{G}}(\varepsilon, \delta)$ is the minimum integer $m$ for which there exists an ($\varepsilon, \delta$)-estimator.*

The query complexity depends on both the value distribution class $\mathcal{C}$ and the reward function class $\mathcal{G}$. In this work, we will consider two natural classes of value distributions: (1) $\mathcal{C}_{\mathrm{ALL}}$, the set of all distributions supported on $[0, 1]$; (2) $\mathcal{C}_{\mathrm{LIP}}$, the set of distributions on $[0, 1]$ whose CDF $F$ is Lipschitz continuous. And we consider two types of reward functions: *monotone* and *right-Lipschitz* (with respect to $\gamma$). Specifically, for any fixed $v \in [0, 1]$, define *projection* $g_v \triangleq g(\cdot, v)$, which is a one dimensional function of $\gamma \in [0, v]$. Let $\mathcal{G}_{\mathrm{MONO}}$ be the set of reward functions $g$ whose projection $g_v$ is weekly increasing (w.r.t.$\gamma$) for all $v \in [0, 1]$. For the Lipschitzness, we define:

**Definition 2.3** (Lipschitzness). *A one dimensional function $f$ is*

- *($L$-)left-Lipschitz, if for all $x, y \in \mathrm{dom}(f)$ with $x \leq y$, $f(y) - f(x) \geq -L(y - x)$.*
- *($L$-)right-Lipschitz, if for all $x, y \in \mathrm{dom}(f)$ with $x \leq y$, $f(y) - f(x) \leq L(y - x)$.*
- *one-sided ($L$-)Lipschitz, if it is either ($L$-)left-Lipschitz or ($L$-)right-Lipshitz.*
- *($L$-)Lipschitz, if it is both left- and right-Lipshitz.*

Let $\mathcal{G}_{\mathrm{RIGHT\text{-}LIP}}$ ($\mathcal{G}_{\mathrm{LIP}}$) be the set of reward functions $g$ whose projection $g_v$ is right-Lipschitz (Lipschitz) for all $v \in [0, 1]$. Monotonicity and right Lipschitzness are natural assumptions of the reward functions. In the above examples, the rewards (quality of the hire, revenue, and the quality of collected data) are all *weakly increasing* with respect to the thresholds (hiring bar, reserve price, and the difficulty of the requirement). For right-Lipschitz functions, one can see Duetting et al. (2023) for some practical examples.

---

[1]This reduction is proved to be without loss of generality in Feng et al. (2021).

## 3 IMPOSSIBILITY RESULT: MONOTONE REWARD FUNCTION AND GENERAL VALUE DISTRIBUTION

In this section, we investigate the query complexity of learning the optimal threshold for general value distributions. We show that even when the reward function $g$ is monotone with respect to both the threshold $\gamma$ and the latent value $v$, an $(\varepsilon, \delta)$-estimator cannot be learned with finitely many queries, even for a constant $\varepsilon = \frac{1}{8}$.

**Theorem 3.1.** *For any $\delta \in (0, 1)$ and $\varepsilon \le \frac{1}{8}$, the query complexity $\mathrm{QC}_{\mathcal{G}_{\mathrm{MONO}}, \mathcal{C}_{\mathrm{ALL}}}(\varepsilon, \delta)$ is infinite.*

**High-Level Ideas** To prove the theorem, we carefully construct a set of pairs of monotone reward function and value distribution $(g_\alpha, F_\alpha) \in \mathcal{G}_{\mathrm{MONO}} \times \mathcal{C}_{\mathrm{ALL}}$, parameterized by $\alpha \in (\frac{1}{2}, \frac{9}{16})$, such that that the utility function $U(\gamma)$ has a unique maximum point at $\gamma^* = \alpha$ and $U(\gamma) < U(\alpha) - \varepsilon$ for any $\gamma \ne \alpha$. To learn an $\varepsilon$-approximately optimal threshold, the learner must find the point $\gamma^* = \alpha$. But there are infinitely many possible values for $\alpha \in (\frac{1}{2}, \frac{9}{16})$, and our construction ensures that the learner cannot determine the exact value of $\alpha$ from the feedback of finitely many queries.

*Proof.* Fix any $\alpha \in (\frac{1}{2}, \frac{9}{16})$. We define value distribution $F_\alpha$ with the following CDF:

$$F_\alpha(v) = \begin{cases} 0 & v \in [0, \frac{1}{2}) \\ \frac{1}{2} - \frac{3}{16v} & v \in [\frac{1}{2}, \alpha) \\ \frac{1}{2} & v \in [\alpha, 1) \\ 1 & v = 1. \end{cases}$$

More specifically, $F_\alpha$ consists of the following four parts: (1) A point mass at $\frac{1}{2}$: $\mathbb{P}(v = \frac{1}{2}) = \frac{1}{8}$; (2) Continuous CDF over the interval $(\frac{1}{2}, \alpha)$: $F_\alpha(x) = \frac{1}{2} - \frac{3}{16x}$; (3) A point mass at $\alpha$: $\mathbb{P}(v = \alpha) = \frac{3}{16\alpha}$; (4) A point mass at 1: $\mathbb{P}(v = 1) = \frac{1}{2}$. Then we define a reward function $g_\alpha$ that is monotone with respect to both the threshold $\gamma$ and the latent value $v$:

$$g_\alpha(\gamma, v) = \begin{cases} \gamma & \gamma \in [0, \alpha), v \in [0, \frac{15}{16}) \\ v - \frac{3}{8} & \gamma \in [0, \alpha), v \in [\frac{15}{16}, 1] \\ \gamma & \gamma \in [\alpha, 1], v \in [0, \frac{15}{16}) \\ v & \gamma \in [\alpha, 1], v \in [\frac{15}{16}, 1]. \end{cases}$$

Now we can compute the expected utility $U_\alpha(\gamma)$ when the learner chooses the threshold $\gamma \in [0, 1]$ for the reward function $g_\alpha$ and value distribution $F_\alpha$. We have

- when $\gamma \in [0, \frac{1}{2}]$, the utility is $U_\alpha(\gamma) = \frac{1}{2}\gamma + \frac{5}{16} < \frac{9}{16}$;
- when $\gamma \in (\frac{1}{2}, \alpha)$, the utility is $U_\alpha(\gamma) = (F(\alpha) - F(\gamma))\gamma + \frac{5}{16} = \frac{1}{2}$;
- when $\gamma = \alpha$, the utility is $U_\alpha(\gamma) = \alpha \cdot \mathbb{P}[v = \alpha] + \frac{1}{2} = \frac{11}{16}$;
- when $\gamma \in (\alpha, 1]$, the utility is $U_\alpha(\gamma) = \frac{1}{2}$.

Therefore, $U_\alpha(\gamma)$ is maximized at $\gamma^* = \alpha$, and for any $\gamma \ne \alpha$, it holds that $U_\alpha(\alpha) - U_\alpha(\gamma) > \frac{1}{8}$. To approximate the optimal utility within $\varepsilon = \frac{1}{8}$ error, the learner must learn the exact value of $\alpha$. However, the following claim shows that when a learner chooses any threshold $\gamma \in [0, 1]$, it only observes censored feedback in $\{\gamma, \frac{5}{8}, 1, 0\}$.

**Claim 3.1.** *When the learner chooses $\gamma \in [0, 1]$, it observes feedback in $\{0, \gamma, \frac{5}{8}, 1\}$.*

*Proof.* We consider two cases. If the learner chooses a threshold $\gamma < \alpha$, the learner only receives feedback in $\{\frac{5}{8}, \gamma, 0\}$ depending on the latent value $v$: (1) if $v \ge \frac{15}{16}$, the learner receives $g_\alpha(\gamma, v) = v - \frac{3}{8} = 1 - \frac{3}{8} = \frac{5}{8}$; (2) if $\gamma \le v_t < \frac{15}{16}$, the learner receives $g_\alpha(\gamma, v) = \gamma$; (3) if $v < \gamma$, the learner receives 0. Similarly, if the learner chooses a threshold $\gamma \ge \alpha$, the learner only receives feedback in $\{0, 1, \gamma\}$: (1) if $v \ge \frac{15}{16}$, the learner receives $g_\alpha(\gamma, v) = v = 1$; (2) if $\gamma \le v < \frac{15}{16}$, the learner receives $g_\alpha(\gamma, v) = \gamma$; (3) if $v < \gamma$, the learner receives 0. This proves the claim. □

Note that the above results holds for all $\alpha \in (\frac{1}{2}, \frac{9}{16})$. By properties of $U_\alpha(\gamma)$ and Claim 3.1, a learner is not able to output the exact value of $\alpha$ using finite queries with feedback only in $\{0, \gamma, \frac{5}{8}, 1\}$ against infinitely many pairs of $(F_\alpha, g_\alpha)$ for $\alpha \in (\frac{1}{2}, \frac{9}{16})$. □

## 4 TIGHT QUERY COMPLEXITY FOR THE LIPSCHITZ CASES

### 4.1 MONOTONE REWARD FUNCTION AND LIPSCHITZ VALUE DISTRIBUTION

The negative result in Section 3 implies that we need more assumptions on the reward function $g$ or the value distribution $F$ for the learner to learn the optimal threshold in finite queries. In this subsection, we keep assuming that $\mathcal{G}$ is the class of monotone functions w.r.t. $\gamma$ and further assume that $\mathcal{C}$ is the class of Lipschitz distributions. We argue that the monotonicity of $g$ and the Lipschitzness of $F$ guarantee that the gap between the expected utility of two thresholds can be bounded by the gap between their CDFs.

**Lemma 4.1.** *With $g \in \mathcal{G}_{\mathrm{MONO}}$ and $F \in \mathcal{C}_{\mathrm{LIP}}$, the expected utility function $U$ satisfies that $U(\gamma_2) - U(\gamma_1) \geq -(F(\gamma_2) - F(\gamma_1))$ for any $0 \leq \gamma_1 \leq \gamma_2 \leq 1$.*

Then we show that $\tilde{O}(\frac{1}{\varepsilon^3})$ queries are enough to learn the optimal threshold. And this can be achieved even if the learner does not know the Lipschitz constant $L$ of the value distribution $F$.

**Theorem 4.1.** *For $\mathcal{G}_{\mathrm{MONO}}$ and $\mathcal{C}_{\mathrm{LIP}}$, we have*

$$\mathrm{QC}_{\mathcal{G}_{\mathrm{MONO}},\mathcal{C}_{\mathrm{LIP}}}(\varepsilon, \delta) \leq O\left(\frac{1}{\varepsilon^3} \log \frac{L}{\varepsilon} \log \frac{\log \frac{L}{\varepsilon}}{\varepsilon\delta}\right).$$

*Proof.* If the learner chooses a threshold $\gamma$ and makes $m$ queries with the same threshold $\gamma$, it will observe $m$ i.i.d. samples $b_1(\gamma), b_2(\gamma), \ldots, b_m(\gamma)$. Here we use $b_i(\gamma)$ to denote the random variable $b(\gamma, v_i)$ where $v_i \sim F$. Let $G_\gamma$ be the CDF of the distribution of $b_i(\gamma)$. Let $\hat{G}_\gamma$ be the CDF of the empirical distribution: $\hat{G}_\gamma(x) = \sum_{i=1}^{m} \mathbf{1}_{b_i(\gamma) \leq x}$. By the DKW inequality (Lemma B.1), if the number of queries $m$ reaches $O(\frac{1}{\varepsilon^2} \log \frac{1}{\delta'})$, then with probability at least $1 - \delta'$ we have

$$\left|G_\gamma(x) - \hat{G}_\gamma(x)\right| \leq \varepsilon, \quad \forall x \in \mathbb{R}.$$

By Tonelli's theorem, $U(\gamma) = \mathbb{E}_{b(\gamma) \sim G_\gamma}[b(\gamma)] = \int_0^1 \mathbb{P}[b(\gamma) > t]dt = \int_0^1 \left(1 - G_\gamma(t)\right)dt$.

Define $\hat{U}(\gamma) = \int_0^1 \left(1 - \hat{G}_\gamma(t)\right)dt$. Then we have

$$\left|U(\gamma) - \hat{U}(\gamma)\right| = \left|\int_0^1 \left(G_\gamma(t) - \hat{G}_\gamma(t)\right)dt\right| \leq \varepsilon. \tag{3}$$

This means that, after $O(\frac{1}{\varepsilon^2} \log \frac{1}{\delta'})$ queries at the same point $\gamma$, we can learn the expected utility of threshold $\gamma$ in $\varepsilon$ additive error with probability at least $1 - \delta'$.

Next, we will adaptively build a discretization set $\Gamma_A \subset [0, 1]$ *without knowing the Lipschitz constant* $L$ such that (1) $\Gamma_A = \{x_1, x_2, \cdots, x_{|\Gamma_A|}\}$ where $0 = x_1 < x_2 < \cdots < x_{|\Gamma_A|} = 1$ and $|\Gamma_A| = \Theta(\frac{1}{\varepsilon})$; (2) $\frac{\varepsilon}{3} < F(x_{i+1}) - F(x_i) < \varepsilon$ for $i = 1, 2, \cdots, |\Gamma_A| - 1$.

By Chernoff bound, we know that $O(\frac{1}{\varepsilon^2} \log \frac{1}{\delta})$ queries are sufficient to learn $F(x)$ with $\frac{\varepsilon}{9}$ additive error for any $x \in [0, 1]$. At step 1, let $\Gamma_A = \{0\}$. At step $n > 1$, assuming $\Gamma_A = \{x_1, x_2, ..., x_{n-1}\}$ and the estimations of corresponding CDF $\hat{F}(x_1), \hat{F}(x_2), ..., \hat{F}(x_{n-1})$ are computed. Let $x_{n-1} = \max \Gamma_A$. Then we use binary search to find the next element $x_n$ satisfying $\frac{2\varepsilon}{3} - \frac{\varepsilon}{9} < \hat{F}(x_n) - \hat{F}(x_{n-1}) < \frac{2\varepsilon}{3} + \frac{\varepsilon}{9}$ and hence $\frac{\varepsilon}{3} < F(x_n) - F(x_{n-1}) < \varepsilon$. The binary search works because the empirical CDF function $\hat{F}$ is monotone. Note that we can always find such $x_n$ within $O(\log \frac{L}{\varepsilon})$ points because of the Lipschitzness. Overall, to build $\Gamma_A$, we need to estimate $n = O(\log \frac{L}{\varepsilon} \cdot |\Gamma_A|) = O(\frac{1}{\varepsilon} \log \frac{L}{\varepsilon})$ points of the value distribution. By union bound, to successfully build $\Gamma_A$ with probability $1 - \frac{\delta}{2}$, the total query complexity is $O(n \cdot \frac{1}{\varepsilon^2} \log \frac{n}{\delta}) = O(\frac{1}{\varepsilon^3} \log \frac{L}{\varepsilon} \log \frac{\log \frac{L}{\varepsilon}}{\varepsilon\delta})$.

Once $\Gamma_A$ is built, we make $O(\frac{1}{\varepsilon^2} \log \frac{2|\Gamma_A|}{\delta})$ for each threshold $\gamma \in \Gamma_A$. By the union bound, we have with probability at least $1 - |\Gamma_A| \cdot \frac{\delta}{2|\Gamma_A|} = 1 - \frac{\delta}{2}$, the estimate $\hat{U}(\gamma)$ satisfies $|\hat{U}(\gamma) - U(\gamma)| \leq \varepsilon$ for every threshold $\gamma \in \Gamma_A$. Since $|\Gamma_A| = O(\frac{1}{\varepsilon})$, we need $|\Gamma_A| \cdot O(\frac{1}{\varepsilon^2} \log \frac{|\Gamma_A|}{\delta}) = O(\frac{1}{\varepsilon^3} \log \frac{1}{\varepsilon\delta})$

queries in total. Then, let $\gamma^* = \arg\max_{\gamma \in [0,1]} U(\gamma)$ be the optimal threshold in $[0, 1]$ for $U(\gamma)$ and let $\hat{\gamma}^* = \arg\max_{\gamma \in \Gamma_A} \hat{U}(\gamma)$ be the optimal threshold in $\Gamma_A$ for $\hat{U}(\gamma)$. And let $\gamma_r \in \Gamma_A$ such that $\gamma_{r-1} < \gamma^* \leq \gamma_r$, so $0 \leq F(\gamma_r) - F(\gamma^*) < \varepsilon$. Then, we have the following chain of inequalities:

$$U(\hat{\gamma}^*) \overset{\text{Eq. (3)}}{\geq} \hat{U}(\hat{\gamma}^*) - \varepsilon \overset{\text{Definition of } \hat{\gamma}^*}{\geq} \hat{U}(\gamma_r) - \varepsilon \overset{\text{Eq. (3)}}{\geq} U(\gamma_r) - 2\varepsilon$$

$$\overset{\text{Lemma 4.1}}{\geq} (U(\gamma^*) - \varepsilon) - 2\varepsilon \geq U(\gamma^*) - 3\varepsilon.$$

We conclude that $\hat{\gamma}^*$ is a $3\varepsilon$-optimal threshold with probability $1 - \delta$. $\qquad\square$

## 4.2 RIGHT-LIPSCHITZ REWARD FUNCTION AND GENERAL VALUE DISTRIBUTION

In this section, we first show that the expected utility function is right-Lipschitz when $\mathcal{G}$ is the class of right-Lipschitz continuous reward function and $\mathcal{C}$ is the class of general distribution.

**Lemma 4.2.** *Suppose reward function $g \in \mathcal{G}_{\text{RIGHT-LIP}}$, then the expected reward function $U$ is right-Lipschitz continuous.*

Then assuming that we know the Lipschitz constant $L$, we can apply similar discretization method in Theorem 4.1 to provide an upper bound.

**Theorem 4.2.** *For the class of right-Lipschitz reward functions $\mathcal{G}_{\text{RIGHT-LIP}}$ and general distributions $\mathcal{C}_{\text{ALL}}$, $\varepsilon > 0$, $\delta \in [0, 1]$, we have*

$$\text{QC}_{\mathcal{G}_{\text{LIP}}, \mathcal{C}_{\text{LIP}}}(\varepsilon, \delta) \leq \text{QC}_{\mathcal{G}_{\text{RIGHT-LIP}}, \mathcal{C}_{\text{ALL}}}(\varepsilon, \delta) \leq O\left(\frac{L}{\varepsilon^3} \log \frac{L}{\varepsilon\delta}\right).$$

## 4.3 LOWER BOUND

In this section, we prove that even if $\mathcal{G}$ is the class of reward functions that are both Lipschitz and monotone w.r.t. $\gamma$ and $\mathcal{C}$ is the class of Lipschitz distributions, $\Omega(\frac{1}{\varepsilon^3})$ queries are needed to learn the optimal utility within $\varepsilon$ error. This is a uniformly matching lower bound for all the upper bounds in this paper.

**Theorem 4.3.** *For $\mathcal{G} = \mathcal{G}_{\text{LIP}} \cap \mathcal{G}_{\text{MONO}}$, $\mathcal{C}_{\text{LIP}}$, we have*

$$\text{QC}_{\mathcal{G}, \mathcal{C}_{\text{LIP}}}(\varepsilon, \delta) \geq \Omega\left(\frac{1}{\varepsilon^3} + \frac{1}{\varepsilon^2} \log \frac{1}{\delta}\right).$$

**High-level ideas** To prove the theorem, we construct a Lipschitz value distribution and carefully perturb it on a $O(\varepsilon)$ interval. The base value distribution leads to any $\gamma \in [\frac{1}{3}, \frac{1}{2}]$ being the optimal threshold. However, the perturbed distribution leads to a unique optimal threshold $\gamma^* \in [\frac{1}{3}, \frac{1}{2}]$. To learn an $\varepsilon$-approximately optimal threshold, the learner needs to distinguish the base distribution and $O(\frac{1}{\varepsilon})$ perturbed distributions. It can be proved that $\Omega(\frac{1}{\varepsilon^2} \log \frac{1}{\delta})$ queries are needed to distinguish the base distribution and one perturbed distribution, which leads to our desired lower bound. This idea is significantly different from previous works that constructed discrete distributions (e.g., Kleinberg & Leighton (2003); Leme et al. (2023b)).

*Proof.* Let $g(\gamma, v) = \gamma$ for any $\gamma, v \in [0, 1]$ such that $v \geq \gamma$. It is not difficult to verify that $g$ is Lipschitz continuous and monotone w.r.t. $\gamma$. In this case, the expected utility can be written in a simple form $U(\gamma) = \gamma(1 - F(\gamma))$.

Consider the following Lipschitz continuous distribution $F_0$:

$$F_0(v) = \begin{cases} \frac{3}{4}v & v \in [0, \frac{1}{3}) \\ 1 - \frac{1}{4v} & v \in [\frac{1}{3}, \frac{1}{2}] \\ v & v \in (\frac{1}{2}, 1]. \end{cases}$$

This distribution leads to the following expected utility function:

$$U_0(\gamma) = \begin{cases} \gamma(1 - \frac{3}{4}\gamma) & \gamma \in [0, \frac{1}{3}) \\ \gamma(1 - (1 - \frac{1}{4\gamma})) = \frac{1}{4} & \gamma \in [\frac{1}{3}, \frac{1}{2}] \\ \gamma(1 - \gamma) & \gamma \in (\frac{1}{2}, 1]. \end{cases}$$

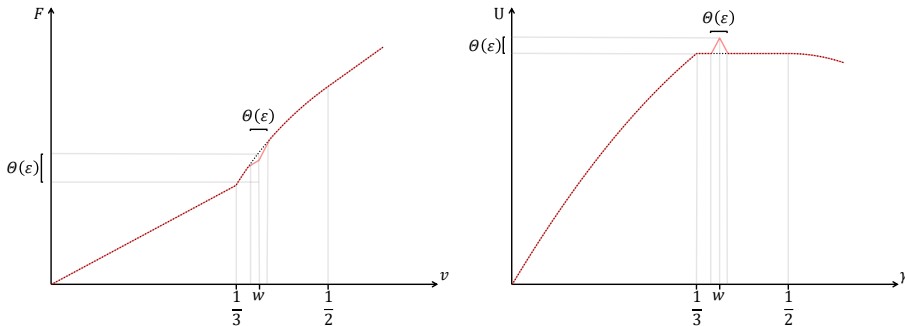

Figure 1: Left: The base distribution $F$ (black, dotted) and the perturbed distribution $F_{w,\gamma}$ (red solid), which moves one unit of mass from interval $[w - 3\varepsilon, w]$ to interval $[w, w + 3\varepsilon]$. Right: The corresponding qualitative plots of $\gamma \mapsto U(\gamma)$ (black, dotted) and $\gamma \mapsto U_{w,\gamma}(\gamma)$ (red, solid).

Because $\gamma(1 - \frac{3}{4}\gamma)$ is increasing on interval $[0, \frac{1}{3})$ and $\gamma(1 - \gamma)$ is decreasing on interval $(\frac{1}{2}, 1]$, $U_0(\gamma) < U_0(\frac{1}{3}) = \frac{1}{4}$ when $\gamma \in [0, \frac{1}{3})$ and $U_0(\gamma) < U_0(\frac{1}{2}) = \frac{1}{4}$ when $\gamma \in (\frac{1}{2}, 1]$. In other words, the expected utility function reaches maximum value if and only if $\gamma \in [\frac{1}{3}, \frac{1}{2}]$. There is a "plateau" on the utility curve. (See Figure 1.)

Next, we perturb the distribution $F_0$ to obtain another distribution $F_{\omega,\varepsilon}$, whose expected utility function will only have one maximum point rather than the interval $[\frac{1}{3}, \frac{1}{2}]$. So to estimate the optimal utility, the learner must distinguish $F_0$ and a class of perturbed distributions. However, by carefully designing the perturbation, the difference between $F_0$ and perturbed distributions is small enough to lead to the desired lower bound.

Let $\Xi = \{(w, \varepsilon) \in [0, 1]^2, w - 3\varepsilon \geq \frac{1}{3}, w + 3\varepsilon \leq \frac{1}{2}\}$. For any $(w, \varepsilon) \in \Xi$, let $h_{w,\varepsilon}(v) = \mathbf{1}_{v \in (w, w+3\varepsilon]} - \mathbf{1}_{v \in [w-3\varepsilon, w)}$. Notice that $F_0$ has the following probability distribution function $f_0$:

$$f_0(v) = \begin{cases} \frac{3}{4} & v \in [0, \frac{1}{3}) \\ \frac{1}{4v^2} & v \in [\frac{1}{3}, \frac{1}{2}] \\ 1 & v \in (\frac{1}{2}, 1]. \end{cases}$$

So $f_0(v) \geq 1$ when $v \in [\frac{1}{3}, \frac{1}{2}]$. Define $f_{w,\varepsilon}(v) = f_0(v) + h_{w,\varepsilon}(v)$, then we have $f_{w,\varepsilon}(v) \geq 0$ for any $v \in [0, 1]$. And $\int_{v=0}^1 f_{w,\varepsilon}(v) = 1$, so $f_{w,\varepsilon}$ is a valid probability density function. Let $F_{w,\varepsilon}$ be the corresponding cumulative distribution function. Note that $F_{w,\varepsilon}$ is $L$-Lipschitz for any constant $L \geq \frac{13}{4}$. By definition, $F_{w,\varepsilon}(v) = \int_0^v f_{\omega,\varepsilon}(t)dt = \int_0^v (f_0(t) + h_{w,\varepsilon}(t))dt = F_0(v) + \int_0^v h_{w,\varepsilon}(t)dt$.

**Claim 4.1.** *(See proof in Appendix C.4) The CDF function $F_{w,\varepsilon}$ is*

$$F_{w,\varepsilon}(v) = \begin{cases} F_0(v) - (v - w + 3\varepsilon) & v \in [w - 3\varepsilon, w) \\ F_0(v) - (w + 3\varepsilon - v) & v \in [w, w + 3\varepsilon] \\ F_0(v) & \text{otherwise.} \end{cases}$$

Claim 4.1 means that $F_{w,\varepsilon}$ and $F_0$ are nearly the same except on the $6\varepsilon$-long interval $[w-3\varepsilon, w+3\varepsilon]$.

Let $U_{w,\varepsilon}(\gamma)$ be the expected utility function when the agent's value distribution is $F_{w,\varepsilon}$.

$$U_{w,\varepsilon}(\gamma) - U_0(\gamma) = \gamma\big(F_0(\gamma) - F_{w,\varepsilon}(\gamma)\big) = \begin{cases} \gamma(\gamma - (w - 3\varepsilon)) & \gamma \in [w - 3\varepsilon, w) \\ \gamma(w + 3\varepsilon - \gamma) & \gamma \in [w, w + 3\varepsilon] \\ 0 & \text{otherwise.} \end{cases}$$

$U_{w,\varepsilon}$ has a unique maximum point at $\gamma = w$, with $U_{w,\varepsilon}(w) = U_0(w) + w\varepsilon = \frac{1}{4} + 3w\varepsilon > \frac{1}{4} + \varepsilon$, because $\gamma(\gamma - (w - 3\varepsilon))$ is increasing in $[w - 3\varepsilon, w)$ and $\gamma(w + 3\varepsilon - \gamma)$ is decreasing in $[w, w + 3\varepsilon]$.

Let $w_i = \frac{1}{3} + 3(2i - 1)\varepsilon, i \in \{1, 2, ..., n = \lfloor \frac{1}{36\varepsilon} \rfloor\}$. Note that $(w_i, \varepsilon) \in \Xi$ for all $i \in [n]$. It can be proved that distinguishing distributions $F_0$ and $F_{w_i,\varepsilon}$ requires $\Omega(\frac{1}{\varepsilon^2} \log \frac{1}{\delta})$ queries in the interval $[w_i - 3\varepsilon, w_i + 3\varepsilon]$ (see Appendix C.5). Queries not in $[w_i - 3\varepsilon, w_i + 3\varepsilon]$ do not help to distinguish $F_0$ and $F_{w_i,\varepsilon}$ because their CDFs are the same outside of the range $[w_i - 3\varepsilon, w_i + 3\varepsilon]$.

**Lemma 4.3.** *For any $i \in [n]$, $\Omega(\frac{1}{\varepsilon^2} \log \frac{1}{\delta})$ queries in interval $[w_i - 3\varepsilon, w_i + 3\varepsilon]$ are needed to distinguish $F_0$ and $F_{w_i, \varepsilon}$.*

Now consider a setting where the underlying value distribution is $F_{w_i, \varepsilon}$ for $i$ uniformly drawn from $\{1, 2, \ldots, n\}$. Finding out the optimal utility and the corresponding threshold is equivalent to finding out the underlying value distribution. On one side, fixing $\delta$, for each $i \in [n]$ we need $\Omega(\frac{1}{\varepsilon^2})$ queries to distinguish $F_{w_i, \varepsilon}$ and $F_0$ and all these queries must lie in the interval $I_i = [w_i - 3\varepsilon, w_i + 3\varepsilon]$. Since these intervals $\{I_i\}_{i \in [n]}$ are disjoint, and the learner must make $\Omega(\frac{1}{\varepsilon^2})$ queries in each interval $I_i$, this leads to $n \cdot \Omega(\frac{1}{\varepsilon^2}) = \Omega(\frac{1}{\varepsilon^3})$ queries in total. On the other side, for any $\delta \in (0, 1)$, $\Omega(\frac{1}{\varepsilon^2} \log \frac{1}{\delta})$ are needed to distinguish the underlying value distribution and other distributions with probability at least $1 - \delta$. Combining these two lower bounds, $\Omega(\frac{1}{\varepsilon^3} + \frac{1}{\varepsilon^2} \log \frac{1}{\delta})$ queries are needed to learn the optimal threshold in $\varepsilon$ additive error with probability at least $1 - \delta$. $\qquad \square$

**Corollary 4.1.** $\mathrm{QC}_{\mathcal{G}_{\mathrm{LIP}}, \mathcal{C}_{\mathrm{LIP}}}(\varepsilon, \delta) \geq \Omega(\frac{1}{\varepsilon^3} + \frac{1}{\varepsilon^2} \log \frac{1}{\delta})$, $\mathrm{QC}_{\mathcal{G}_{\mathrm{MONO}}, \mathcal{C}_{\mathrm{LIP}}}(\varepsilon, \delta) \geq \Omega(\frac{1}{\varepsilon^3} + \frac{1}{\varepsilon^2} \log \frac{1}{\delta})$.

## 5 ONLINE LEARNING

Previous sections studied the threshold learning problem in an offline query complexity model. In this section, we consider an (adversarial) online learning model and show that the threshold learning problem in this setting can be solved with a tight $\Theta(T^{2/3})$ regret in the Lipschitz case.

**Adversarial online threshold learning** The learner and the agent interact for $T$ rounds. At each round $t \in [T]$, the learner selects a threshold $\gamma_t \in [0, 1]$ and the agent realizes a value $v_t \sim F_t \in \mathcal{C}$. The learner then observes reward $b_t = b_t(\gamma_t, v_t) = g_t(\gamma_t, v_t) \mathbf{1}_{v_t \geq \gamma_t}$ as in Eq. (1), which depends on the unknown reward function $g_t \in \mathcal{G}$ and the value $v_t$. Unlike the query complexity model where the learner only cares about the quality of the final output threshold $\hat{\gamma}$, here, the learner cares about the total reward gained during the $T$ rounds: $\sum_{t=1}^{T} b_t(\gamma_t, v_t)$. We compare this total reward against the best total reward the learner could have obtained using some fixed threshold in hindsight. In other words, we measure the performance of the learner's algorithm $\mathcal{A}$ by its *regret*:

$$\mathrm{Reg}_T^{\mathcal{C}, \mathcal{G}}(\mathcal{A}) = \sup_{\gamma \in [0,1]} \mathbb{E}\left[\sum_{t=1}^{T} b_t(\gamma, v_t) - \sum_{t=1}^{T} b_t(\gamma_t, v_t)\right]. \tag{4}$$

Unlike the query complexity model where the value distribution $F$ and reward function $g$ are fixed, in this online learning model we allow them to change over time. Moreover, they can be controlled by an *adaptive adversary* who, given classes $\mathcal{C}$ and $\mathcal{G}$, at each round $t$ can arbitrarily choose an $F_t \in \mathcal{C}$ and a $g_t \in \mathcal{G}$ based on the history $(\gamma_1, v_1, \ldots, \gamma_{t-1}, v_{t-1})$.

We have shown in Section 3 that learning an $\varepsilon$-optimal threshold is impossible for monotone reward functions $\mathcal{G}_{\mathrm{MONO}}$ and general distributions $\mathcal{C}_{\mathrm{ALL}}$. Similarly, it is impossible to obtain $o(1)$ regret in this case. So, we focus on the two cases with finite query complexity: $\mathcal{S}_1 = (\mathcal{G}_{\mathrm{RIGHT\text{-}LIP}}, \mathcal{C}_{\mathrm{ALL}})$, $\mathcal{S}_2 = (\mathcal{G}_{\mathrm{MONO}}, \mathcal{C}_{\mathrm{LIP}})$. The following theorem shows that, for those two cases, there exists a learning algorithm that achieves $\Theta(T^{2/3})$ regret, and this bound is tight.

**Theorem 5.1.** *For the adversarial online threshold learning problem, there exists an algorithm $\mathcal{A}$ that, for both cases $\mathcal{S}_1 = (\mathcal{G}_{\mathrm{RIGHT\text{-}LIP}}, \mathcal{C}_{\mathrm{ALL}})$ and $\mathcal{S}_2 = (\mathcal{G}_{\mathrm{MONO}}, \mathcal{C}_{\mathrm{LIP}})$, achieves regret*

$$\mathrm{Reg}_T^{\mathcal{S}_i}(\mathcal{A}) \leq O(T^{2/3} L^{1/3}).$$

*And for any algorithm $\mathcal{A}$, there exists a fixed reward function $g \in \mathcal{G}_{\mathrm{LIP}} \cap \mathcal{G}_{\mathrm{MONO}}$ and a fixed distribution $F \in \mathcal{C}_{\mathrm{LIP}}$ for which the regret of $\mathcal{A}$ is at least $\mathrm{Reg}_T^{g; F}(\mathcal{A}) \geq \Omega(T^{2/3})$.*

The proof idea is as follows. The lower bound $\Omega(T^{2/3})$ follows from the $\Omega(\frac{1}{\varepsilon^3})$ query complexity lower bound in Theorem 4.3 by a standard online-to-batch conversion. To prove the upper bound $O(T^{2/3} L^{1/3})$, we note that under the two environments $\mathcal{S}_1, \mathcal{S}_2$, the expected reward function $U_t(\gamma) = \mathbb{E}_{v_t \sim F_t}[b(\gamma, v_t)]$ is one-sided Lipschitz in $\gamma$. Therefore, we can treat the problem as a continuous-arm one-sided Lipschitz bandit problem, where each threshold $\gamma \in [0, 1]$ is an arm. This problem can be solved by discretizing the arm set and running a no-regret bandit algorithm for a finite arm set, e.g., Poly INF (Audibert & Bubeck, 2010). See details in Appendix D.

ACKNOWLEDGMENTS

We acknowledge the financial support from the National Natural Science Foundation of China (NSFC) (No.62172012). We thank all anonymous reviewers for their helpful feedback.

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

## A  ADDITIONAL RELATED WORKS

For the online setting, our work is most related to the Lipschitz bandit problem, which was first studied by Agrawal (1995). Once we have Lipschitzness, there is a standard discretization method to get the desired upper bound and it is widely used in online settings. See Magureanu et al. (2014); Kleinberg et al. (2019); Haghtalab et al. (2022). The upper bound of our online results follows this standard method, but the lower bound relies on our offline results and is different from previous continuous-armed Lipschitz bandit problems. Several recent works study multi-armed bandit problems with censored feedback. For example, Abernethy et al. (2016) study a bandit problem where the learner obtains a reward of 1 if the realized sample associated with the pulled arm is larger than an exogenously given threshold. Verma et al. (2019) study a resource allocation problem where the learner obtains reward only if the resources allocated to an arm exceed a certain threshold. In Guinet et al. (2022), the reward of each arm is missing with some probability at every pull. These models are significantly different from ours, hence the results are not comparable.

Censored/Truncated data are also widely studied in statistical analysis. For example, a classical problem is truncated linear regression, which has remained a challenge since the early works of Tobin (1958); Amemiya (1973); Breen (1996). Recently, Daskalakis et al. (2019) provided a computationally and statistically efficient solution to this problem. Statistics problems with known or unknown truncated sets have received much attention recently (e.g. (Daskalakis et al., 2018; Kontonis et al., 2019)). For more knowledge of this field, the reader can turn to the textbook of Little & Rubin (2020). While this line of work studies passive learning settings where the censored dataset is given to the data analyst exogenously, we consider an active learning setting where the learner can choose how to censor each data point, and how to do that optimally.

## B  BASIC MATH

### B.1  DKW INEQUALITY

The following well-known concentration inequality will be used in our proofs.

**Lemma B.1** (Dvoretzky–Kiefer–Wolfowitz (DKW) inequality (Dvoretzky et al., 1956; Massart, 1990))**.** *Let $X_1, X_2, \ldots, X_n$ be $n$ real-valued i.i.d random variables with cumulative distribution function $F$. Let $F_n = \sum_{i=1}^{n} \mathbf{1}_{X_i \leq x}$ be the associated empirical distribution function. Then for all $\varepsilon > 0$,*

$$\mathbb{P}\left[\sup_{x \in \mathbb{R}} \left(F_n(x) - F(x)\right) > \varepsilon\right] < 2e^{-2n\varepsilon^2}.$$

### B.2  PROPERTIES OF HELLINGER DISTANCE

Then we review some useful properties of the Hellinger distance and total variation distance. First, the Hellinger distance gives upper bounds on the total variation distance:

**Fact B.1.** *Let $D_1$, $D_2$ be two distribution on $\mathcal{X}$. Their total variance distance and Hellinger distance are $d_{\mathrm{TV}}(D_1, D_2)$ and $d_{\mathrm{H}}(D_1, D_2)$ respectively. We have*

$$1 - d_{\mathrm{TV}}^2(D_1, D_2) \geq (1 - d_{\mathrm{H}}^2(D_1, D_2))^2.$$

The total variation distance has the following well-known property that upper bounds the difference between the expected values of a function on two distributions:

**Fact B.2.** *For any function $h : \mathcal{X} \to [0, 1]$, $|\mathbb{E}_{x \sim D_1}[h(x)] - \mathbb{E}_{x \sim D_2}[h(x)]| \leq d_{\mathrm{TV}}(D_1, D_2)$.*

Second, we use the following lemma to upper bound the squared Hellinger distance between two distributions that are close to each other. We use $D$ to specifically denote discrete distributions. We slightly abuse the notation by using $D$ to denote of PDF of distribution $D$.

**Lemma B.2.** *Let $D_1$, $D_2$ be two distribution on $\mathcal{X}$ satisfying $1 - \varepsilon \leq \frac{D_2(x)}{D_1(x)} \leq 1 + \varepsilon$ for all $x \in \mathcal{X}$. Then $d_{\mathrm{H}}^2(D_1, D_2) \leq \frac{1}{2}\varepsilon^2$.*

*Proof.* By definition,

$$d_{\mathrm{H}}^2(D_1, D_2) = \frac{1}{2} \sum_{x \in \mathcal{X}} (\sqrt{D_1(x)} - \sqrt{D_2(x)})^2 = \frac{1}{2} \sum_{x \in \mathcal{X}} D_1(x)\left(1 - \sqrt{\frac{D_2(x)}{D_1(x)}}\right).$$

If $D_2(x) < D_1(x)$, then we have $1 - \sqrt{\frac{D_2(x)}{D_1(x)}} \le (1 - \sqrt{1-\varepsilon})^2 \le \left(1 - (1-\varepsilon)\right)^2 = \varepsilon^2$. If $D_2(x) \ge D_1(x)$, we have $1 - \sqrt{\frac{D_2(x)}{D_1(x)}} \le (\sqrt{1+\varepsilon} - 1)^2 \le \left((1+\varepsilon-1)\right)^2 = \varepsilon^2$. Combining these two cases, we have

$$d_{\mathrm{H}}^2(D_1, D_2) \le \frac{1}{2} \sum_{x \in \mathcal{X}} D_1(x)\varepsilon^2 = \frac{1}{2}\varepsilon^2.$$

$\square$

Finally, let $D^{\otimes m}$ denote the empirical distribution of $T$ i.i.d samples from $D$, namely, the product of $m$ independent $D$ distributions. We have the following lemma relates $d_{\mathrm{H}}^2(D_1^{\otimes m}, D_2^{\otimes m})$ with $d_{\mathrm{H}}^2(D_1, D_2)$.

**Lemma B.3.** *(Lee (2020))* $d_{\mathrm{H}}^2(D_1^{\otimes m}, D_2^{\otimes m}) = 1 - (1 - d_{\mathrm{H}(D_1,D_2)}^2)^m \le m \cdot d_{\mathrm{H}}^2(D_1, D_2).$

### B.3 DISTINGUISHING DISTRIBUTIONS

Let $D_1, D_2$ be two distributions over a discrete space $\mathcal{X}$. A distribution $D_i$ is chosen uniformly from the set $\{D_1, D_2\}$. Then we are given $m$ samples from $D_i$ and want to distinguish whether the distribution is $D_1$ or $D_2$. It is known that at least $m = \Omega(\frac{1}{d_{\mathrm{H}}^2(D_1,D_2)} \log \frac{1}{\delta})$ samples are needed to guess correctly with probability at least $1 - \delta$, no matter how we guess. Formally we have

**Lemma B.4.** *Let $j \in \{1, 2\}$ be the index of the distribution we guess based on the samples. The probability of making a mistake when distinguishing $D_1$ and $D_2$ using $m$ samples, namely $\Pr[j \ne i] = \frac{1}{2}\Pr[j = 2|i = 1] + \frac{1}{2}\Pr[j = 1|i = 2]$, is at least*

$$\Pr[j \ne i] \ge \frac{1}{4}\left(1 - d_{\mathrm{H}}^2(D_1, D_2)\right)^{2m} \ge \frac{1}{4}e^{-4m d_{\mathrm{H}}^2(D_1,D_2)},$$

*if $d_{\mathrm{H}}^2(D_1, D_2) \le \frac{1}{2}$. The inequality implies that, in order to achieve $\Pr[j \ne i] \le \delta$, we must have $m \ge \frac{1}{4d_{\mathrm{H}}^2(D_1,D_2)} \log \frac{1}{4\delta}$.*

*Proof.* The draw of $m$ samples from $D_1$ or $D_2$ is equivalent to the draw of one sample from $D_1^{\otimes m}$ or $D_2^{\otimes m}$. Given one sample from $D_1^{\otimes m}$ or $D_2^{\otimes m}$, the probability of making one mistake when guessing the distribution is at least

$$\begin{aligned}
\Pr[j \ne i] &= \frac{1}{2}\Pr[j = 2|i = 1] + \frac{1}{2}\Pr[j = 1|i = 2] \\
&= \frac{1}{2}\left(1 - \Pr[j = 1|i = 1] + \frac{1}{2}\Pr[j = 1|i = 2]\right) \\
&= \frac{1}{2} - \frac{1}{2}(\Pr[j = 1|i = 2] - \Pr[j = 1]|i = 1]) \\
&= \frac{1}{2} - \frac{1}{2}\left(\mathbb{E}_{D_1^{\otimes m}}[\mathbf{1}_{j=1}] - \mathbb{E}_{D_2^{\otimes m}}[\mathbf{1}_{j=1}]\right) \\
by\,Fact\,B.2 \quad &\ge \frac{1}{2} - \frac{1}{2}d_{\mathrm{TV}}\left(D_1^{\otimes m}, D_2^{\otimes m}\right).
\end{aligned} \tag{5}$$

Then we upper bound $d_{\mathrm{TV}}(D_1^{\otimes m}, D_2^{\otimes m})$ to prove the lemma. According to Fact B.1 and Lemma B.3, we have

$$1 - d_{\mathrm{TV}}^2\left(D_1^{\otimes m}, D_2^{\otimes m}\right) \ge \left(1 - d_{\mathrm{H}}^2\left(D_1^{\otimes m}, D_2^{\otimes m}\right)^2\right) = \left(1 - d_{\mathrm{H}}^2(D_1, D_2)\right)^{2m}.$$

Since

$$1 - d_{\mathrm{TV}}^2\Big(D_1^{\otimes m}, D_2^{\otimes m}\Big) = \Big(1 + d_{\mathrm{TV}}\Big(D_1^{\otimes m}, D_2^{\otimes m}\Big)\Big)\Big(1 - d_{\mathrm{TV}}\Big(D_1^{\otimes m}, D_2^{\otimes m}\Big)\Big)$$
$$\leq 2\Big(1 - d_{\mathrm{TV}}\Big(D_1^{\otimes m}, D_2^{\otimes m}\Big)\Big),$$

we have

$$1 - d_{\mathrm{TV}}\Big(D_1^{\otimes m}, D_2^{\otimes m}\Big) \geq \frac{1}{2}\big(1 - d_{\mathrm{H}}^2(D_1, D_2)\big)^{2m}.$$

Plugging into Eq. (5), we have

$$\Pr[j \neq i] \geq \frac{1}{4}\big(1 - d_{\mathrm{H}}^2(D_1, D_2)\big)^{2m}.$$

When $d_{\mathrm{H}}^2(D_1, D_2) < \frac{1}{2}$, the inequality $1 - x \geq e^{-2x}$ for all $x \in (0, \frac{1}{2})$ concludes that

$$\Pr[j \neq i] \geq \frac{1}{4}e^{-4m d_{\mathrm{H}}^2(D_1, D_2)}.$$

$\square$

## C  MISSING PROOFS FROM SECTION 4

### C.1  PROOF OF LEMMA 4.1

*Proof.* The distribution $F$ is weakly differentiable because it is Lipschitz. Let $f$ be the weak derivative of $F$. We have $f(v) \leq L$ for all $v \in [0, 1]$ because of Lipschitzness. We rewrite the expected utility $U(\gamma) = \mathbb{E}_{v \sim F}\big[g(\gamma, v) \cdot \mathbf{1}_{v \geq \gamma}\big]$ as $\int_{v=\gamma}^{1} g(\gamma, v)f(v)dv$. Then for any $0 \leq \gamma_1 \leq \gamma_2 \leq 1$,

$$U(\gamma_2) - U(\gamma_1) = \int_{v=\gamma_2}^{1} g(\gamma_2, v)f(v)dv - \int_{v=\gamma_1}^{1} g(\gamma_1, v)f(v)dv$$
$$= \int_{v=\gamma_2}^{1} \big(g(\gamma_2, v) - g(\gamma_1, v)\big)f(v)dv - \int_{v=\gamma_1}^{\gamma_2} g(\gamma_1, v)f(v)dv$$
$$\geq 0 - \int_{v=\gamma_1}^{\gamma_2} g(\gamma_1, v)f(v)dv \quad \geq \quad -(F(\gamma_2) - F(\gamma_1))$$

where the first inequality holds because $g$ is monotone in $\gamma$, the second inequality holds because $g(\gamma, v) \leq 1$. $\square$

### C.2  PROOF OF LEMMA 4.2

*Proof.* For any $0 \leq \gamma_1 \leq \gamma_2 \leq 1$,

$$U(\gamma_2) - U(\gamma_1) = \mathbb{E}_{v \sim F}[g(\gamma_2, v) \cdot \mathbf{1}_{v \geq \gamma_2}] - \mathbb{E}_{v \sim F}[g(\gamma_1, v) \cdot \mathbf{1}_{v \geq \gamma_1}]$$
$$= \mathbb{E}_{v \sim F}[\big(g(\gamma_2, v) - g(\gamma_1, v)\big) \cdot \mathbf{1}_{v \geq \gamma_2}] - \mathbb{E}_{v \sim F}[g(\gamma_1, v) \cdot \mathbf{1}_{\gamma_1 \leq v < \gamma_2}]$$
$$\leq \mathbb{E}_{v \sim F}[\big(g(\gamma_2, v) - g(\gamma_1, v)\big) \cdot \mathbf{1}_{v \geq \gamma_2}] \quad \leq \quad L(\gamma_2 - \gamma_1)$$

where the first inequality holds because $g(\gamma, v) \geq 0$, the second inequality holds because the projection $g_v$ is right-Lipschitz continuous and the expectation $\mathbb{E}_{v \sim F}[\mathbf{1}_{v \geq \gamma_2}] \leq 1$. $\square$

### C.3  PROOF OF THEOREM 4.2

*Proof.* Because $\mathcal{C}_{\mathrm{LIP}} \subset \mathcal{C}_{\mathrm{ALL}}$, we have $\mathrm{QC}_{\mathcal{G}_{\mathrm{LIP}}, \mathcal{C}_{\mathrm{LIP}}}(\varepsilon, \delta) \leq \mathrm{QC}_{\mathcal{G}_{\mathrm{LIP}}, \mathcal{C}_{\mathrm{ALL}}}(\varepsilon, \delta)$ for all $\varepsilon > 0$, $\delta \in (0, 1)$. For any $\varepsilon > 0$, let's consider all the multiples of $\frac{\varepsilon}{L}$ in $[0, 1]$, $\Gamma \triangleq \{\frac{k\varepsilon}{L} : k \in \mathbb{N}, k \leq \frac{L}{\varepsilon}\}$. We make $O(\frac{1}{\varepsilon^2}\log\frac{|\Gamma|}{\delta})$ queries for each threshold $\gamma = \frac{k\varepsilon}{L}$ in $\Gamma$. By a union bound, we have with probability at least $1 - |\Gamma| \cdot \frac{\delta}{|\Gamma|} = 1 - \delta$, the estimate $\hat{U}(\gamma)$ satisfies $|\hat{U}(\gamma) - U(\gamma)| \leq \varepsilon$ for every threshold in $\Gamma$. Since $|\Gamma| = O(\frac{L}{\varepsilon})$, this uses $|\Gamma| \cdot O(\frac{1}{\varepsilon^2}\log\frac{|\Gamma|}{\delta}) = O(\frac{L}{\varepsilon^3}\log\frac{L}{\varepsilon\delta})$ queries in total.

Let $\gamma^* \in \arg\max_{\gamma \in [0,1]} U(\gamma)$ be an optimal threshold. And let $\hat{\gamma}^* \in \arg\max_{\gamma \in \Gamma}$ be the optimal threshold on the discretized set. And let $\gamma_l = \frac{\varepsilon}{L} \lfloor \frac{L\gamma^*}{\varepsilon} \rfloor$ respectively be the multiples of $\frac{\varepsilon}{L}$ closest to the left. Then we have $\gamma_l \in \Gamma$ and $0 < \gamma^* - \gamma_l < \varepsilon$. Since the reward function g is right-$L$-Lipschitz-continuous,

$$
\begin{aligned}
U(\hat{\gamma}^*) &\geq \hat{U}(\hat{\gamma}^*) - \varepsilon \geq \hat{U}(\gamma_l) - \varepsilon \\
&\geq U(\gamma_l) - 2\varepsilon \\
&\geq U(\gamma^*) - 2\varepsilon - L(\gamma^* - \gamma_l) \\
&\geq U(\gamma^*) - 3\varepsilon
\end{aligned}
$$

where the first and third inequality holds because Eq. (3), the second inequality holds because the selection of $\hat{\gamma}^*$, the fourth inequality holds because of Lemma 4.2. □

## C.4 PROOF OF CLAIM 4.1

*Proof.* For $v < w - 3\varepsilon$ and $v > w + 3\varepsilon$, $\int_{t=0}^{v} h_{w,\varepsilon}(t)dt = 0$ because $h_{w,\varepsilon}(t) = 0$ when $t < w - 3\varepsilon$ or $t > w + 3\varepsilon$ and $\int_{t=w-3\varepsilon}^{w+3\varepsilon} h_{w,\varepsilon}(t)dt = 3\varepsilon - 3\varepsilon = 0$. Therefore, for $v < w - 3\varepsilon$ and $v > w + 3\varepsilon$, $F_{w,\varepsilon}(v) = F_0(v)$. For any $v \in [w - 3\varepsilon, w)$,

$$
F_{w,\varepsilon}(v) - F_0(v) = \int_{t=w-3\varepsilon}^{v} h_{w,\varepsilon}(t)dt = -(v - w + 3\varepsilon).
$$

And for any $v \in [w, w + 3\varepsilon]$,

$$
F_{w,\varepsilon}(v) - F_0(v) = \int_{w-3\varepsilon}^{v} h_{w,\varepsilon}(t)dt = \int_{w-3\varepsilon}^{w+3\varepsilon} h_{w,\varepsilon}(t)dt - \int_{v}^{w+3\varepsilon} h_{w,\varepsilon}(t)dt = -(w + 3\varepsilon - v).
$$

□

## C.5 PROOF OF LEMMA 4.3

*Proof.* When the learner sets different thresholds $\gamma$ and the value distribution is $F_0$, the samples come from different distributions $G_\gamma$. Similarly, when the learner sets different thresholds $\gamma$ and the value distribution is $F_{w,\varepsilon}$, assume that the samples come from different distributions $G_\gamma^{w,\varepsilon}$.

In order to distinguish $F_0$ and $F_{w,\varepsilon}$, the learner must at least find a threshold $\gamma$ that it is able to distinguish $G_\gamma$ and $G_\gamma^{w,\varepsilon}$.

Given $g(\gamma, v) = \gamma \cdot \mathbf{1}_{v \geq \gamma}$, $G_\gamma$ is a Bernoulli distribution that for all $X \sim G_\gamma$, $\Pr[X = 0] = F_0(\gamma)$ and $\Pr[X = \gamma] = 1 - F_0(\gamma)$. And $G_\gamma^{w,\varepsilon}$ is also a Bernoulli distribution that for all $Y \sim G_\gamma^{w,\varepsilon}$, $\Pr[Y = 0] = F_{w,\varepsilon}(\gamma)$ and $\Pr[Y = \gamma] = 1 - F_{w,\varepsilon}(\gamma)$.

Recall that

$$
F_{w,\varepsilon}(v) = \begin{cases} F_0(v) - (v - w + 3\varepsilon) & v \in [w - 3\varepsilon, w) \\ F_0(v) - (w + 3\varepsilon - v) & v \in [w, w + 3\varepsilon] \\ F_0(v) & \text{otherwise} \end{cases}
$$

When $\gamma < w - 3\varepsilon$ or $\gamma > w + 3\varepsilon$, $G_\gamma$ and $G_\gamma^{w,\varepsilon}$ are the same Bernoulli distribution. The learner can't distinguish them. When $\gamma \in [w - 3\varepsilon, w + 3\varepsilon]$, we have $F_0(\gamma) \geq F_{w,\varepsilon}(\gamma) \geq F_0(\gamma) - 3\varepsilon$ and $F_0(\frac{1}{2}) \geq F_0(\gamma) \geq F_0(\frac{1}{3})$. Recall $F_0(\frac{1}{3}) = \frac{1}{4}$ and $F_0(\frac{1}{2}) = \frac{1}{2}$. Then we get

$$
1 \geq \frac{\Pr[Y = 0]}{\Pr[X = 0]} = \frac{F_{w,\varepsilon}(\gamma)}{F_0(\gamma)} \geq \frac{F_0(\gamma) - 3\varepsilon}{F_0(\gamma)} = 1 - \frac{3\varepsilon}{F_0(\gamma)} \geq 1 - \frac{3\varepsilon}{F_0(\frac{1}{3})} = 1 - 12\varepsilon.
$$

and

$$
1 \leq \frac{\Pr[Y = \gamma]}{\Pr[X = \gamma]} = \frac{1 - F_{w,\varepsilon}(\gamma)}{1 - F_0(\gamma)} \leq \frac{1 - F_0(\gamma) + 3\varepsilon}{1 - F_0(\gamma)} = 1 + \frac{3\varepsilon}{1 - F_0(\gamma)} \leq 1 + \frac{3\varepsilon}{1 - F_0(\frac{1}{2})} = 1 + 6\varepsilon.
$$

According to Lemma B.2, we have $d_{\mathrm{H}}^2(G_\gamma, G_\gamma^{w,\varepsilon}) \leq \frac{1}{2}(12\varepsilon)^2 \leq 72\varepsilon^2$. Then we know from Lemma B.4 that $\Omega(\frac{1}{\varepsilon^2} \log \frac{1}{\delta})$ samples are needed to distinguish $G_\gamma$ and $G_\gamma^{w,\varepsilon}$ with probability at least $1 - \delta$.

In other words, the learner must at least find a threshold $\gamma \in [w - 3\varepsilon, w + 3\varepsilon]$ and do at least $\Omega(\frac{1}{\varepsilon^2} \log \frac{1}{\delta})$ queries at the same threshold $\gamma$ to distinguish $F_0$ and $F_{w,\varepsilon}$ with probability at least $1 - \delta$. □

## D  PROOF OF THEOREM 5.1

*Proof.* First, because $v_t \sim F_t$ is independent of $\gamma_t$, we have $\mathbb{E}_{v_t \sim F_t}[b_t(\gamma_t, v_t)] = U_t(\gamma_t)$, and we can rewrite the regret as

$$\sup_{\gamma \in [0,1]} \mathbb{E}\left[\sum_{t=1}^T b_t(\gamma, v_t) - \sum_{t=1}^T b_t(\gamma_t, v_t)\right] = \sup_{\gamma \in [0,1]}\left\{\mathbb{E}\left[\sum_{t=1}^T U_t(\gamma)\right] - \mathbb{E}\left[\sum_{t=1}^T U_t(\gamma_t)\right]\right\},$$

where the expectation on the right-hand-side is only over the randomness of algorithm $\mathcal{A}$ but not $v_t$.

We treat the online learning problem as a continuous-arm adversarial bandit problem, where each threshold $\gamma \in [0,1]$ is an arm. According to Lemma 4.1 and Lemma 4.2, in all the three environments $\mathcal{S}_1, \mathcal{S}_2, \mathcal{S}_3$ in the theorem the expected utility function $U_t(\gamma) = \mathbb{E}_{v_t \sim F_t}[b_t(\gamma, v_t)]$ is one-sided Lipschitz in $\gamma$. W.l.o.g, assume that $U_t$ is right-Lipschitz. Let's discretize the arm space $[0,1]$ uniformly with interval length $\frac{\varepsilon}{L}$, obtaining a finite set of arms $\Gamma = \{0, \frac{\varepsilon}{L}, \frac{2\varepsilon}{L}, ...\}$ with $|\Gamma| \leq \frac{L}{\varepsilon} + 1$. Let $\gamma^* \in \arg\max_{\gamma \in [0,1]} \mathbb{E}\left[\sum_{t=1}^T U_t(\gamma)\right]$ be an optimal threshold in the interval $[0,1]$ (for the expected sum of utility functions). And let $\hat{\gamma}^* \in \arg\max_{\gamma \in \Gamma} \mathbb{E}\left[\sum_{t=1}^T U_t(\gamma)\right]$ be an optimal threshold in the discretized set $\Gamma$. And let $\hat{\gamma}_l \in \Gamma$ be the largest multiple of $\varepsilon$ that does not exceed $\gamma^*$. Clearly, $\gamma^* - \hat{\gamma}_l \leq \frac{\varepsilon}{L}$. Because every $U_t$ is right-Lipschitz, we have

$$\mathbb{E}\left[\sum_{t=1}^T U_t(\gamma^*)\right] - \mathbb{E}\left[\sum_{t=1}^T U_t(\hat{\gamma}_l)\right] \leq \sum_{t=1}^T L(\gamma^* - \hat{\gamma}_l) \leq T\varepsilon.$$

This implies that the optimal threshold $\hat{\gamma}^*$ in $\Gamma$ satisfies

$$\mathbb{E}\left[\sum_{t=1}^T U_t(\hat{\gamma}^*)\right] \geq \mathbb{E}\left[\sum_{t=1}^T U_t(\hat{\gamma}_l)\right] \geq \mathbb{E}\left[\sum_{t=1}^T U_t(\gamma^*)\right] - T\varepsilon.$$

Recall that the Poly INF algorithm (Theorem 11 of (Audibert & Bubeck, 2010)) is an adversarial multi-armed bandit algorithm with $O(\sqrt{TK})$ regret when running on an arm set of size $K$. If we run that algorithm on the arm set $\Gamma$, and let $\varepsilon = (\frac{L}{T})^{1/3}$, then we get a total expected utility of at least

$$\mathbb{E}\left[\sum_{t=1}^T U_t(\gamma_t)\right] \geq \mathbb{E}\left[\sum_{t=1}^T U_t(\hat{\gamma}^*)\right] - O(\sqrt{T|\Gamma|})$$

$$\geq \mathbb{E}\left[\sum_{t=1}^T U_t(\gamma^*)\right] - T\varepsilon - O\left(\sqrt{T\frac{L}{\varepsilon}}\right)$$

$$= \mathbb{E}\left[\sum_{t=1}^T U_t(\gamma^*)\right] - O(T^{2/3}L^{1/3}).$$

So, the regret is at most $O(T^{2/3}L^{1/3})$. □

## E  EXPERIMENTAL RESULTS

In this section, we provide some simple experiments to verify our theoretical results.

### E.1 UPPER BOUND

We consider two toy examples. The first example is $g(\gamma, v) = \gamma$ if $\gamma < \frac{1}{3}$ otherwise $g(\gamma, v) = v$ and the value distribution is the uniform distribution on $[0, 1]$, which corresponds to the monotone reward function and Lipschitz value distribution. The second example is $g(\gamma, v) = \gamma$ and the value distribution is a point distribution that all the mass is on $v = \frac{1}{3}$, which corresponds to the Lipschitz reward function and general distribution case (Theorem 4.2). In our experiment, we first fix the number of queries to be $K^3$ where $K = 100 + 3i$ for all integer $1 \leq i \leq 33$, i.e. we choose $K$ from $[100, 200]$. Therefore, the algorithm for the upper bound should output errors smaller than the predetermined loss $\frac{1}{K}$. The following figure shows the relationship between the loss and the number of queries. The empirical loss curve is under the predetermined loss curve, which verifies our upper bound results.

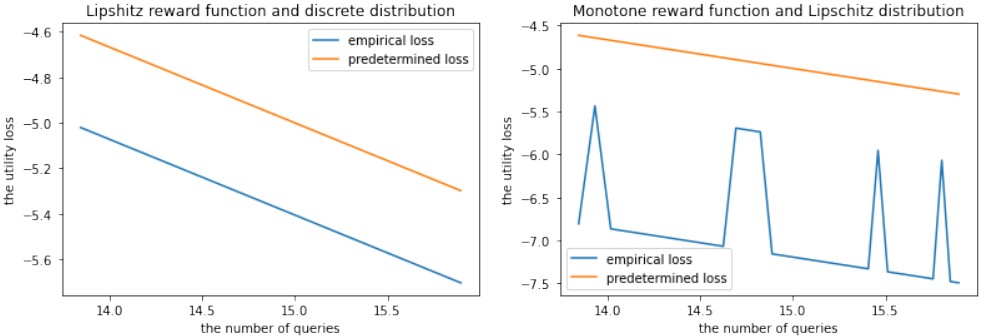

Figure 2: Loss curves under different examples. The orange line: the predetermined loss curve. The blue line: the empirical loss curve. All variables are in logarithmic form.

### E.2 LOWER BOUND

In this section, we provide experimental results to verify our lower bound result (Theorem 4.3). We consider the example we provided in the proof of Theorem 4.3 where $g(\gamma, v) = \gamma$ and the value distribution is a "hard distribution" (see the left part of Fig. 3). For $\varepsilon \in \{\frac{1}{400}, \frac{1}{500}, \frac{1}{600}\}$, we run the algorithm in Theorem 4.1 to determine the minimum number $n$ of queries that are necessary to learn the optimal threshold with $\varepsilon$ additive error. Due to randomness, we repeat 10 times for each $\varepsilon$. At each round, we compute $\frac{\ln n}{\ln \frac{1}{\varepsilon}}$ and find it converging to 3, which verifies our lower bound result.

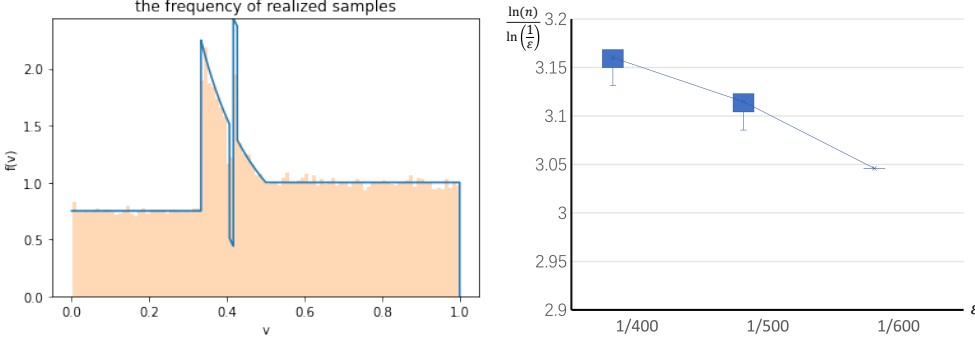

Figure 3: Left: The blue curve is the probability distribution function. The orange part is the frequency of realized samples. Right: The box plot when $\varepsilon \in \{\frac{1}{400}, \frac{1}{500}, \frac{1}{600}\}$. The horizontal axe represents $\varepsilon$. The vertical axe represents the logarithmic ratio $\frac{\ln n}{\ln \frac{1}{\varepsilon}}$.

