# OpenReview forum: "Learning Thresholds with Latent Values and Censored Feedback"
_ICLR.cc/2024/Conference — ICLR 2024 poster_

### Official Review · Reviewer_MPBo · 2023-10-13

**Soundness:** 3 good
**Presentation:** 3 good
**Contribution:** 2 fair
**Rating:** 6
**Confidence:** 4

**Summary:**

This paper studies the sample complexity of active threshold learning when samples are censored. The authors then apply their offline results to provide a tight regret guarantee for the online version of the problem.

The model is as follows: there is a latent variable $v$ in $[0,1]$ sampled according to an unknown distribution and a reward function $g$ that maps (threshold, latent variable) pairs in $[0,1]$. The learner repeatedly and adaptively queries the value of $g$ on specific thresholds (for i.i.d. realizations of the latent variable) and obtains censored feedback, i.e., if the threshold is larger than the latent variable, it does not observe anything. The goal of the learner is to get an $(\varepsilon,\delta)$ approximation of the best threshold, using as few (adaptive) samples as possible.

The results are as follows:
in the general case, the sample complexity is infinite (Theorem 3.1.)
when function $g$ is Lipshitz or the latent variable distribution has Lipshitz CDF, then $\tilde O(\tfrac{1}{\varepsilon^3})$ samples are enough (Theorems 4.1 and 4.2)
these bounds are tight up to poly-logarithmic terms (Theorem 4.3)
these results immediately imply a $\Theta(T^{2/3})$ minimax regret regime for the online learning version of the problem

**Strengths:**

Censored data are interesting as they naturally arise from applications and have been studied both in the sample complexity and online learning literature. The paper is well written and the proofs in the main body are easy to understand. The fact that the authors provide a complete picture of the problem is compelling.

**Weaknesses:**

My main concern with the paper, which motivates my low score, is given by the technical contribution of the paper, as all the theoretical results are not too surprising and build on known techniques.
- the needle in a haystack phenomenon is known and has been used before in pricing context, see e.g., [1] “A Regret Analysis of Bilateral Trade" EC, 2021
- Theorems 4.1 and 4.2 are not surprising: if some regularity or smoothness assumptions are made then the objective becomes Lipshitz in the threshold, and an epsilon grid on the threshold space immediately yields the desired result see. e.g., [2] “Bandits and Experts in Metric Spaces” J ACM, 2019.
- Theorem 4.3 is the more involved result but is not too surprising: the construction entails a family of distributions with $\Theta(\tfrac 1{\varepsilon})$ candidate optimal thresholds where each candidate needs to be evaluated $\Omega(\tfrac 1{\varepsilon^2})$ times. This has already been done, e.g., in the $\Omega(T^{2/3})$ lower bounds in [3] “The Value of Knowing a Demand Curve: Bounds on Regret for Online Posted-Price Auctions” FOCS 2003.

The problem studied, and the techniques used, are closely related to Lipshitz bandits [2], pricing [3] and bilateral trade [1]. Please consider a more thorough comparison with the already known results and techniques there.

%%%%
I thank the authors for the insightful comments and the submission revision. Overall, I am convinced that the technical contribution --although deeply inspired by past works- is novel enough to raise my score slightly.
I particularly appreciated that the authors added some comments and pointers to the literature before the proofs to highlight this.

**Questions:**

- It seems like Theorem 5.1. could be a Corollary of [2] (for the upper bound, once Lipshitzness in the threshold has been established) and [3] (for the lower bound). Is it correct?

---

> ### Author Response · Authors · 2023-11-17
> **Thanks for your comments!**
>
> Thanks for your insightful comments with citations. We sincerely appreciate your time in reading the paper and providing useful references. Our point-to-point responses to your comments are given below.
>
> > 1. The needle in a haystack phenomenon is known and has been used before in the pricing context.
>
> Thank you for this insightful comment! We fully agree with the reviewer that the "needle in a haystack" phenomenon is a known concept. In response to this valuable feedback, we have added additional references in our revised paper to emphasize this phenomenon.  However, we wish to clarify that it is more like a high-level idea rather than a panacea that can be used without careful consideration. In [1], they use this idea to give a linear regret lower bound (Theorem 4.4). In [4], they also use this idea to prove matching lower bounds and it is already cited in our original paper. Nevertheless, it is important to note that the specific setups they considered differ from ours, resulting in a distinct concrete construction. In essence, our paper's concrete construction stands independently and does not rely on previous works, highlighting the inherent difficulty of the problem we address.
>
> > 2. Theorems 4.1 and 4.2 are not surprising: if some regularity or smoothness assumptions are made then the objective becomes Lipschitz in the threshold, and an epsilon grid on the threshold space immediately yields the desired result.
>
> Thank you for this perceptive comment! We acknowledge that proving upper bounds by constructing an epsilon grid under the Lipschitz assumption is a standard approach.  We included these results primarily for the completeness of our paper. As mentioned in our introduction, the main technical contribution and novelty of this paper lies in the novel and challenging **impossibility result**, along with a **matching lower bound**.
>
> >3. Theorem 4.3 is the more involved result but is not too surprising: the construction entails a family of distributions with $\Theta(\frac{1}{\epsilon})$ candidate optimal thresholds where each candidate needs to be evaluated $\Omega(\frac{1}{\epsilon^2})$ times.
>
> Thank you for this thoughtful comment! We wish to clarify that **our lower bound is built on the assumption that the value distribution is Lipschitz**. Lipschitzness is a double-edged sword that allows a standard solution on the upper bound side but leads to difficulty in the construction of matching lower bound, which we believe is novel in our paper. In contrast to previous works, such as the construction in [3], which built a family of discrete distributions with support on $V = \\{\frac{1}{2}, \frac{1}{2} + \epsilon, \frac{1}{2} + 2\epsilon, . . . , 1 − \epsilon, 1\\}$, and recent work [5], which constructed a family of discrete distributions with support on $V=\\{\frac{1}{2}+ 4\epsilon, \frac{1}{2} + 8\epsilon, …, \frac{3}{4} − 4\epsilon, \frac{3}{4}\\}$, our lower bound result constructs a family of Lipschitz continuous distributions on the interval [0,1].
>
> > 4. It seems like Theorem 5.1. could be a Corollary of  [2] (for the upper bound, once Lipshitzness in the threshold has been established) and [3] (for the lower bound). Is it correct?
>
> Thank you for this interesting question. Not precisely. While the upper bound in Theorem 5.1 does leverage standard techniques from the Lipschitz bandit literature (e.g., [5]), the lower bound, as we mentioned above, does not align with the conventional approach found in previous works, which typically involves constructing a family of discrete distributions. Instead, our unique contribution lies in the ability to construct a family of Lipschitz continuous distributions to establish the lower bound, which is novel and more challenging.
>
> In conclusion, we express our sincere appreciation for the insightful comments accompanied by citations. We believe that our paper's primary contribution, the lower-bound technique, shares high-level similarities (e.g., the idea of “needle in a haystack”) with previous works while exhibiting distinctive and concrete differences, and it makes our lower bound construction challenging and novel.
>
>
> Reference
>
> [1] Cesa-Bianchi N, Cesari T R, Colomboni R, et al. A regret analysis of bilateral trade[C]//Proceedings of the 22nd ACM Conference on Economics and Computation.
>
> [2] Kleinberg R, Slivkins A, Upfal E. Bandits and experts in metric spaces[J]. Journal of the ACM (JACM), 2019, 66(4): 1-77.
>
> [3] Kleinberg R, Leighton T. The value of knowing a demand curve: Bounds on regret for online posted-price auctions[C]//44th Annual IEEE Symposium on Foundations of Computer Science, 2003.
>
> [4] Leme R P, Sivan B, Teng Y, et al. Pricing query complexity of revenue maximization[C]//Proceedings of the 2023 Annual ACM-SIAM Symposium on Discrete Algorithms (SODA).
>
> [5] Cesa-Bianchi N, Cesari T, Colomboni R, et al. The Role of Transparency in Repeated First-Price Auctions with Unknown Valuations[J]. arXiv preprint arXiv:2307.09478, 2023.

---

### Official Review · Reviewer_Z8bt · 2023-10-28

**Soundness:** 3 good
**Presentation:** 3 good
**Contribution:** 3 good
**Rating:** 5
**Confidence:** 2

**Summary:**

This paper considers a new setting of actively learning threshold in latent space with censored feedback, where the rewards can be observed only when the threshold is lower than or equal to the unknown latent value. The reward function g is defined over both the proposed threshold and latent value.  They proved that the query complexity can be infinitely large even when the reward function is monotone with respect to the threshold and the value. When adding assumptions of Lipschitz CDF, they proved tight query complexity up to logarithmic factors. They also extended to the online learning setting, related it to continuous-armed Lipschitz bandit, with theoretical results.

**Strengths:**

- Clear motivation examples for the proposed setting
- Provided high-level ideas and intuition for proofs, which helps readers understand the theorems
- Provided the query complexity lower and upper bound for both Lipschitz and general reward distributions. Table 1 summarizes the results and is very clear.
- Link to continuous-armed Lipschitz bandit in an online learning setting is interesting, with theoretical results.

**Weaknesses:**

- It is not clear how the theoretical results provided in Table 1 and Section 5 (online learning) compare with related/previous work. In related work, the authors mentioned a few closely related works, it would be helpful to provide a detailed discussion or comparison when applicable.
- No experimental results. It would be good to show empirical results which verify the theoretical results (see below question 1). Additionally, it would be even better if the toy example could correspond to the motivation example provided in the paper.

Happy to raise the score if my main concerns are addressed.

**Questions:**

- Can you provide a few toy examples as running examples to explain the setup and theorems (Table 1)? e.g. concrete distributions under different assumptions, possible (\epsilon, \delta)-estimator and corresponding query complexity bounds, etc.
- For the online setting, the proposed setup links to continuous-arm one-sided Lipschitz bandit problem. can you add the related work about this? Also, as mentioned in the weakness, adding a comparison to the existing results in bandits literature is needed.

---

> ### Author Response · Authors · 2023-11-17
> **Thanks for your comments!**
>
> Thanks for your constructive comments and suggestions, they are exceedingly helpful to improve our paper. We have carefully incorporated them in the revised paper. In the following, your comments are first stated and then followed by our point-by-point responses.
>
> > 1. Can you provide a few toy examples as running examples to explain the setup and theorems (Table 1)? e.g. concrete distributions under different assumptions, possible $(\epsilon, \delta)$-estimator and corresponding query complexity bounds, etc.
>
> Thank you for this meaningful question. Certainly. We illustrate the setup and theorems using the auction with a reserve price as a running example. In this scenario, the seller, serving as the learner, aims to learn the optimal reserve price for an item. The buyer, acting as the agent, possesses a private value for the item drawn from an unknown distribution and is willing to pay g(\gamma, v) when the reserve price is r, and the value is v. If the function is defined as $g(\gamma, v) = \gamma$ for $v < 1/3$ and $v$ for $v > 1/3$, and the value distribution follows a uniform distribution, the reward function is monotone but not Lipschitz, while the value distribution is Lipschitz. This corresponds to Theorem 4.1. On the other hand, if $g(\gamma, v) = \gamma$, the reward function is Lipschitz, aligning with Theorem 4.2. Moreover, the example corresponding to Theorem 4.3 is included in the proof, where $g(\gamma,v)=\gamma$ and the value distribution is a “hard” distribution.
>
> > 2. It would be good to show empirical results that verify the theoretical results. Additionally, it would be even better if the toy example could correspond to the motivation example provided in the paper.
>
> Thank you for this helpful comment! As you suggested, we have performed experiments within the mentioned examples which indeed can correspond to the motivation example provided in the paper. For the upper bound, we used the aforementioned two toy examples that fit the conditions in Theorem 4.1 and Theorem 4.2 respectively, and showed the relationship between the empirical loss and the number of queries (Please see [toy example 1](https://github.com/Anonymous-authors6/Submission7000/blob/main/upper_bound_toy_example1.png) and [toy example 2](https://github.com/Anonymous-authors6/Submission7000/blob/main/upper_bound_toy_example2.png)). For the lower bound, we used the example in the proof of Theorem 4.3, where $g(\gamma,v)=\gamma$ and the value distribution is a "hard" distribution (Please see [hard distribution](https://github.com/Anonymous-authors6/Submission7000/blob/main/pdf.png)). We fixed some $\epsilon$ and ran simulations to see the minimum number $n$ of queries that are necessary to learn the optimal threshold within $\epsilon$ additive error. Then we computed the logarithmic ratio $\frac{\ln n}{\ln\frac{1}{\epsilon}}$ and found it converging to 3, which verifies our lower bound $\tilde{\Omega}(\frac{1}{\epsilon^3})$ (Please see [lower bound example](https://github.com/Anonymous-authors6/Submission7000/blob/main/lower_bound_logarithmic_ratio.pdf)). For more details, please refer to Appendix E of the revised paper. For more details, please refer to Appendix E of the revised paper.
>
> >3. It is not clear how the theoretical results provided in Table 1 and Section 5 (online learning) compare with related/previous work. In related work, the authors mentioned a few closely related works, it would be helpful to provide a detailed discussion or comparison when applicable.
>
> Thank you for this instructive comment. Certainly. We have enhanced the discussion in the related work section of the paper to include a more detailed comparison with closely related works, as you suggested. Particularly, **we have added related work about the continuous-arm one-sided Lipschitz bandit problem**.

---

### Official Review · Reviewer_F7cS · 2023-10-30

**Soundness:** 3 good
**Presentation:** 3 good
**Contribution:** 2 fair
**Rating:** 5
**Confidence:** 3

**Summary:**

This paper studies an abstraction of a threshold learning problem that arises in a number of applications such as reserve price learning in single-item auctions, crowd-sourced task allocation/data collection, and theoretical models of hiring. Here, the reward function is specified by a value and a threshold. The threshold is chosen by the learner, the value is then drawn from an unknown distribution, and the learner observes the reward if the value exceeds the threshold.

First, the authors construct the existence of a monotone reward function and a “hard” distribution over values involving point masses that requires an unbounded query complexity to learn the optimal threshold. The authors then give tight query complexity bounds under Lipschitzness assumptions on the reward and value distributions. Finally, an extension to the online setting is studied.

**Strengths:**

The question posed has a clear motivation and is a nice abstraction of questions like reserve price learning. The paper itself is a nice complete contribution, characterizing hardness and giving tight query complexity bounds in tractable cases. The writing is clear and precise. The lower bound constructions are interesting.

**Weaknesses:**

The proof of the main query complexity upper bound in Theorem 4.1 doesn’t seem too surprising. Under conditions on the distribution, concentration bounds allow the true reward at any given threshold point to be sufficiently estimated, and then the learner runs that estimation for a suitable discretization of possible threshold values. The same proof gives the main upper bound results for the other settings as well.

Overall, my main concern is with the originality and novelty of the question being studied. While it is a neat way to abstract away from more specific applications like reserve price learning, and I think the results presented are certainly nice ones, the overall motivation through the examples of reserve pricing, crowdsourcing, and hiring are not enough to convince me that this is a sufficiently novel work for publication in ICLR. Perhaps a more fine-grained structural analysis in terms of practically-motivated properties of the reward function is possible here, which would make the story more compelling.

The assumption that the learner knows the Lipschitz constant of the unknown distribution in the querying algorithm in the proof of Theorem 4.1 seems too strong given the premise of the value distribution being unknown. Perhaps the results could be augmented to remove this assumption, and L is a parameter that must be learned as a part of the querying?

**Questions:**

Are there specific examples of reward functions g that the authors can give that fit the different conditions of their main theorems? Specifically, reward functions that go beyond the simple case of reserve price learning. If such examples could be further motivated in the context of the other two examples (crowdsourced data collection and hiring) presented, that would solidify the premise of the paper.

I think the authors should include all parameters such as the Lipschitz constant L in their query complexity bounds.

The query policy in the proof of Theorem 4.1 requires that the learner knows the Lipschitz constant L of the CDF of the unknown value distribution. So the value distribution is not really fully unknown here. Is there any way to remove the assumption that the learner knows L, or could that lead to exponential/infinite query complexities?

---

> ### Author Response · Authors · 2023-11-17
> **Thanks for your comments!**
>
> Thanks for your constructive comments and suggestions, they are exceedingly helpful in improving our paper. We have carefully incorporated them in the revised paper. In the following, your comments are first stated and then followed by our point-to-point responses.
>
> > 1. The proof of the main query complexity upper bound in Theorem 4.1 doesn’t seem too surprising. Under conditions on the distribution, concentration bounds allow the true reward at any given threshold point to be sufficiently estimated, and then the learner runs that estimation for a suitable discretization of possible threshold values.
>
> Thank you very much for this insightful comment. We acknowledge that the proof of the main query complexity upper bound in Theorem 4.1 employs a standard discretization method. However, it's important to note that the primary technical contribution of our paper lies in presenting novel and challenging impossibility results, along with a matching lower bound, as mentioned in our introduction. Besides we have addressed your question about the Lipschitz constant in the proof of Theorem 4.1, please see our response for the fourth comment, and the details are in our revised paper.  We believe these aspects significantly contribute to the overall significance of our work.
>
> > 2. Are there specific examples of reward functions g that the authors can give that fit the different conditions of their main theorems? If such examples could be further motivated in the context of the other two examples (crowdsourced data collection and hiring) presented, that would solidify the premise of the paper.
>
> Thank you for this valuable question. Here are specific examples of reward functions $g$ that fit different conditions. For g being monotone but not Lipschitz, please refer to the specific g in our impossibility result (Theorem 3.1). For $g$ being Lipschitz but not monotone, consider $g(\gamma, v) = v\gamma + (1 - v)v$, which is not monotone with respect to $v$. For g being both monotone and Lipschitz, consider $g(\gamma, v) = \gamma$, $g(\gamma, v) = v$, and $g(\gamma, v) = c\gamma + (1 - c)v$ for some constant $c$.
>
> The primary focus of our paper is addressing the challenge of learning the optimal threshold when faced with an **unknown** reward function $g$. In the context of the other two examples (crowdsourced data collection and hiring), it is more practical to assume that the employer or taskmaster does not know the specific form of $g$. Nevertheless, we offer interpretations for these examples. In the crowdsourced data example, the reward function $g(\gamma, v) = \gamma h(v)$ can be interpreted as follows. Here, the threshold corresponds to the normalized number of questions (ranging from 0 to 1) in a single task, while the value corresponds to the maximum number of questions the agent is willing to answer. In practical terms, an agent willing to engage with more questions may be inclined to provide more detailed information for each question. Consequently, the reward for each question becomes a monotone function with respect to $v$, denoted as $h(v)$. Thus, the overall reward function takes the form $g(\gamma, v) = \gamma h(v)$. In the hiring example, $g(\gamma, v) = c\gamma + (1 - c)v$ could represent the reward being between the ability bar (threshold) and the employee's actual ability, reflecting the real-world scenario where individuals may not fully utilize their potential.
>
> > 3. I think the authors should include all parameters such as the Lipschitz constant $L$ in their query complexity bounds.
>
> Thank you for this instructive comment. We have included the Lipschitz constant $L$ in our query complexity bounds in our revised paper. Apart from $L$, the only other parameters considered in our analysis are $\epsilon$ and $\delta$, which we also included. Your attention to detail is appreciated.
>
> > 4. Is there any way to remove the assumption in Theorem 4.1 that the learner knows L, or could that lead to exponential/infinite query complexities?
>
> Thanks for this insightful question! Our current sample complexity result in Theorem 4.1 can be extended to the setting where the learner does not know the Lipschitz constant. We have included the result and proof in the Appendix D of the revised paper. The high-level idea is as follows. Instead of constructing a uniform-grid discretization set that depends on $L$, we can adaptively construct a discretization set where the expected rewards of adjacent two points differ at most $\epsilon$. The new query complexity upper bound is $O(\frac{1}{\epsilon^3}\log\frac{L}{\epsilon}\log\frac{\log\frac{L}{\epsilon}}{\epsilon\delta})$, which improves the $\frac{L}{\epsilon^3}$ to $\frac{1}{\epsilon^3}$ while has an additional $\log L$ terms compared with the case that we know $L$. We hope our response can address your main concerns and further verify the technical contribution of our paper.

---

> > ### Comment · Reviewer_F7cS · 2023-11-22
> >
> > I appreciate the detailed response. I will still elect to keep my score (especially in light of reviewer MPBo's comments as well).

---

### Official Review · Reviewer_sD7K · 2023-10-31

**Soundness:** 4 excellent
**Presentation:** 3 good
**Contribution:** 3 good
**Rating:** 8
**Confidence:** 3

**Summary:**

The paper studies the question of learning threshold in latent space. In this problem, we have an unknown reward function $g(\gamma, v)$ and unknown distribution on $v$ and need to pick $\gamma$ that maximizes $E_{v}[g(\gamma, v) \cdot [v \ge \gamma] ]$.

The paper presents the sequence of impossibility and positive results. The first contribution is the proof that query complexity is infinitely large for general monotone functions. The second contribution is the series of positive results when CDF of distribution $v$ is Lipshitz and when g is one-sided Lipshitz with respect to $\gamma$. In both cases, algorithms achieve $O(\epsilon^{-3})$ sample complexity for learning $\epsilon$ approximation.  Additionally, the authors complement this result with matching lower bounds.

**Strengths:**

The paper introduces an important problem that looks very natural and has applications. This is an interesting and practically relevant setting that captures real-world scenarios like setting reserve prices in auctions, difficulty levels in crowdsourcing, and hiring bars in recruiting.

The paper provides a solid theoretical analysis. The authors achieve tight results. The results are technically sound.


The problem formulation and results are clearly explained. The paper is well-written and easy to follow.

Overall, the paper studies an interesting problem and provides nice theoretical results that advance our understanding of learning in latent spaces.

**Weaknesses:**

The applications discussed in the introduction could be expanded with more practical details. For example, how do different auction formats correspond to different functions g?

 The high-level ideas and intuition could be emphasized more.

 Some simple experiments on synthetic data would help support the theoretical results.

**Questions:**

See weaknesses.

---

> ### Author Response · Authors · 2023-11-17
> **Thanks for your comments!**
>
> Thanks for your encouraging words and constructive comments. We sincerely appreciate your time in reading the paper and our point-to-point responses to your comments are given below.
>
> > 1. The applications discussed in the introduction could be expanded with more practical details. For example, how do different auction formats correspond to different functions $g$?
>
> Thank you for pointing out the need for more practical details in the introduction. These details are indeed discussed in Section 2, specifically in Examples 2.1, 2.2, and 2.3. Example 2.1 demonstrates how different auction formats are represented through distinct $g$ functions.
>
> > 2. The high-level ideas and intuition could be emphasized more.
>
> Thanks for your suggestion! We have included the high-level idea of Theorem 4.3 before its proof in our revised paper.
>
> > 3. Some simple experiments on synthetic data would help support the theoretical results.
>
> Thanks for your suggestion! We have conducted experiments on synthetic data with different assumptions, to provide additional support for our theoretical results. For the upper bound, we chose two toy examples that fit the conditions in Theorem 4.1 and Theorem 4.2 respectively, and showed the relationship between the empirical loss and the number of queries (Please see [toy example 1](https://github.com/Anonymous-authors6/Submission7000/blob/main/upper_bound_toy_example1.png) and [toy example 2](https://github.com/Anonymous-authors6/Submission7000/blob/main/upper_bound_toy_example2.png)). For the lower bound, we used the example in the proof of Theorem 4.3, where $g(\gamma,v)=\gamma$ and the value distribution is a "hard" distribution (Please see [hard distribution](https://github.com/Anonymous-authors6/Submission7000/blob/main/pdf.png)). We fixed some $\epsilon$ and ran simulations to see the minimum number $n$ of queries that are necessary to learn the optimal threshold within $\epsilon$ additive error. Then we computed the logarithmic ratio $\frac{\ln n}{\ln\frac{1}{\epsilon}}$ and found it converging to 3, which verifies our lower bound $\tilde{\Omega}(\frac{1}{\epsilon^3})$ (Please see [lower bound example](https://github.com/Anonymous-authors6/Submission7000/blob/main/lower_bound_logarithmic_ratio.pdf)). For more details, please refer to Appendix E of the revised paper.

---

### Meta-Review · Area_Chair_hwmw · 2023-12-14

**Metareview:**

The paper considers the question of active learning a threshold in latent space. The problem is motivated by applications in auctions, crowdsourcing etc. There are many impossibility results as well as positive results. The reviews were mostly positive.

**Justification For Why Not Higher Score:**

NA

**Justification For Why Not Lower Score:**

NA

---

### Decision · Program_Chairs · 2024-01-16

Accept (poster)